# Different Ras isoforms regulate synaptic plasticity in opposite directions

Esperanza López-Merino (ID)[1], Alba Fernández-Rodrigo[1,6], Jessie G Jiang[1], Silvia Gutiérrez-Eisman[1], David Fernández de Sevilla (ID)[2], Alberto Fernández-Medarde (ID)[3,4], Eugenio Santos (ID)[3,4], Carmen Guerra[4,5], Mariano Barbacid[4,5], José A Esteban (ID)[1✉] & Víctor Briz (ID)[1,7✉]

## Abstract

The small GTPase Ras is an intracellular signaling hub required for long-term potentiation (LTP) in the hippocampus and for memory formation. Genetic alterations in Ras signaling (i.e., RASopathies) are linked to cognitive disorders in humans. However, it remains unclear how Ras controls synaptic plasticity, and whether different Ras isoforms play overlapping or distinct roles in neurons. Using genetically modified mice, we show here that H-Ras (the most abundant isoform in the brain) does not promote LTP, but instead long-term depression mediated by metabotropic glutamate receptors (mGluR-LTD). Mechanistically, H-Ras is activated locally in spines during mGluR-LTD via c-Src, and is required to trigger Erk activation and de novo protein synthesis. Furthermore, H-Ras deletion impairs object recognition as well as social and spatial memory. Conversely, K-Ras is the isoform specifically required for LTP. This functional specialization correlates with a differential synaptic distribution of the two isoforms H-Ras and K-Ras, which may have important implications for RASopathies and cognitive function.

**Keywords** mGluR; Hippocampus; Memory; Spine; RASopathies
**Subject Categories** Membranes & Trafficking; Neuroscience; Signal Transduction

## Introduction

Ras proteins are small GTPases that control MAPK/Erk and PI3K/Akt pathways in response to multiple extracellular stimuli, becoming a hub for signal transduction that regulates cell proliferation, survival, and differentiation (Mendoza et al, 2011; Mitin et al, 2005). There are three Ras isoforms encoded by three different genes: H-, N- and K-Ras (Shimizu et al, 1983; Barbacid, 1987), the last of which has two splicing variants, K-Ras4A and K-Ras4B (McGrath et al, 1983; Wang et al, 2001). These isoforms share more than 90% of homology, with the exception of the carboxy-terminal hypervariable region, which undergoes differential posttranslational modifications determining Ras subcellular localization (Omerovic et al, 2007; Castellano and Santos, 2011; Hancock, 2003).

Ras proteins have been extensively studied in the cancer field as oncogenes (Malumbres and Barbacid, 2003), but they are also essential for neuronal functioning and plasticity (Simanshu et al, 2017; Borrie et al, 2017; Krab et al, 2008). In fact, RASopathies are the most common group of neurodevelopmental disorders, affecting one in 1000 newborns (Rauen, 2013; Hebron et al, 2022). This group of disorders, caused by mutations in genes regulating the Ras/MAPK/Erk pathway, encompasses a spectrum of syndromes with overlapping clinical features, including Costello syndrome, Noonan syndrome, synGAP1 syndrome, and neurofibromatosis type 1. Although each syndrome has its unique features, RASopathy patients are similarly affected by congenital heart disease, increased cancer risk, and high prevalence of intellectual disability and autism spectrum disorder (Bustelo et al, 2018). However, whether specific Ras isoforms are responsible for different aspects of neuronal function and cognition remains unknown.

Experience-driven synaptic plasticity is considered fundamental to learning and memory (Bliss and Collingridge, 1993). Although the precise molecular mechanisms underlying the modulation of synaptic strength are still being elucidated, it is clear that alterations in these events contribute to the pathophysiology of many neurodevelopmental disorders (Martin et al, 2000; Forrest et al, 2018). Interestingly, many mouse models of RASopathies show alterations in N-metyl-D-aspartate receptor-dependent long-term potentiation (NMDAR-LTP) and metabotropic glutamate receptor-dependent long-term depression (mGluR-LTD) (Schreiber et al, 2017; Borrie et al, 2017), two prototypical forms of synaptic plasticity in the hippocampus. Ras requirement for NMDAR-LTP is well established (Qin et al, 2005; Ye and Carew, 2010; Zhu et al, 2002), along with its activation in dendritic spines upon LTP induction (Harvey et al, 2008). In contrast, the role of Ras in mGluR-LTD has not yet been addressed directly, despite the fact

[1]Centro de Biología Molecular Severo Ochoa (CSIC-UAM), Madrid, Spain. [2]Facultad de Medicina, Universidad Autónoma de Madrid, Madrid, Spain. [3]Centro de Investigación del Cáncer (CSIC-Universidad de Salamanca), Salamanca, Spain. [4]CIBERONC (Instituto de Salud Carlos III), Madrid, Spain. [5]Centro Nacional de Investigaciones Oncológicas, Madrid, Spain. [6]Present address: Inserm Université de Bordeaux, U1215 Neurocentre Magendie, Bordeaux, France. [7]Present address: Centro Nacional de Sanidad Ambiental (Instituto de Salud Carlos III), Majadahonda, Madrid, Spain. ✉E-mail: jaesteban@cbm.csic.es; victor.briz@isciii.es

that both NMDAR-LTP and mGluR-LTD rely on downstream Erk signaling (Gallagher et al, 2004; Thomas and Huganir, 2004; Patterson et al, 2010; Zhu et al, 2002).

Hence, we aimed at (i) investigating Ras contribution to mGluR-LTD and (ii) dissecting the potential differential role of H- and K-Ras isoforms in synaptic plasticity (to note, K-Ras4B is virtually the only K-Ras isoform expressed in the brain (Pells et al, 1997; Newlaczyl et al, 2017), and will be referred to as K-Ras in this study, for simplicity). Using genetically modified mice and a combination of electrophysiological, live-cell imaging, biochemical and behavioral assays, we now report that H-Ras is specifically required for mGluR-LTD, through a mechanism involving c-Src activity, Erk signaling and de novo protein synthesis, and it is also critical for cognitive function. On the other hand, and contrary to what has been traditionally assumed, K-Ras but not H-Ras is the isoform responsible for NMDAR-LTP. This isoform-specific functional specialization can be explained at least in part by their distinct subcellular localization in neurons, and may be particularly relevant for their implication in neurodevelopmental and cognitive disorders.

## Results

### Ras activity is required for mGluR-LTD via Erk signaling and protein synthesis

To determine whether Ras activity is required for mGluR-LTD, we generated a Ras dominant negative form (Ras-DN), based on the H-Ras sequence and bearing the mutation S17N (Feig and Cooper, 1988; Zhu et al, 2002). GFP-tagged Ras-DN or wild-type Ras (Ras WT) were expressed in CA1 neurons from rat organotypic hippocampal slice cultures for 12–16 h using Sindbis virus. Then, we performed whole-cell voltage-clamp electrophysiological recordings from uninfected (control) and infected CA1 pyramidal neurons while stimulating CA3 Schaffer collaterals. To note, this experimental configuration allows us to examine the contribution of Ras exclusively at the postsynaptic neuron, as only CA1, but not CA3, neurons express the recombinant Ras protein. mGluR-LTD was induced using a paired-pulse protocol consisting of 900 pairs of pulses (with 50 ms inter-stimulus interval) delivered at 1 Hz at Schaffer collaterals in the presence of APV to block NMDAR, as previously described (Huber et al, 2000a). This protocol produced a robust synaptic depression in control neurons, as shown in Fig. 1A (black symbols). Interestingly, we observed a significant reduction in the extent of depression in Ras-DN infected neurons compared to uninfected neurons (Fig. 1A, compare purple and black symbols). In contrast, Ras WT overexpression did not affect mGluR-LTD (Fig. 1A, orange symbols), indicating that the effect of the DN is not a result of Ras overexpression or virus infection. Therefore, these data strongly suggest that postsynaptic Ras activity is required for mGluR-LTD at CA3-CA1 synapses. To note, neither Ras-DN (Fig. EV1A) nor Ras-WT (Fig. EV1B) overexpression altered AMPAR- or NMDAR-mediated basal synaptic transmission, as indicated by simultaneous recordings from infected and neighboring uninfected neurons.

MAPK/Erk and PI3K/Akt pathways are well-known downstream Ras effectors and critical players in mGluR-LTD (Ye and Carew, 2010). Previous work from our laboratory has shown that

PI3K is required presynaptically (but not postsynaptically) for mGluR-LTD in CA3-CA1 synapses (Sánchez-Castillo et al, 2022). On the other hand, Erk activation and postsynaptic de novo protein synthesis are key events for this form of plasticity (Gallagher et al, 2004; Huber et al, 2000). Therefore, we then evaluated whether Ras activity regulates these processes during mGluR-LTD, using the Ras-DN construct. Activation of mGluRs in hippocampal slices with the mGluR1/5 agonist, DHPG ((S)-3,5-dihydroxyphenylgly-cine), produced a significant increase in Erk1/2 phosphorylation, as shown in Fig. 1B (black symbols). We found that Sindbis virus infection per se (GFP expression) increases basal phospho-Erk1/2 levels compared to uninfected slices (Fig. 1B, light green column, "-DHPG"), although this effect was not statistically significant. Despite this basal activation, GFP-expressing neurons were still able to respond to DHPG enhancing Erk1/2 phosphorylation as uninfected hippocampal slices. In contrast, blockade of Ras signaling by Ras-DN overexpression severely impaired the increase in Erk signaling induced by DHPG in the CA1 region of hippocampal slices (Fig. 1B, purple symbols), without affecting basal Erk activity.

To monitor protein synthesis, we employed the surface sensing of translation (SUnSET) assay, a non-radioactive technique based on puromycin labeling (Schmidt et al, 2009). We infected CA1 hippocampal neurons with a mix of Sindbis viruses expressing Ras-DN (labeled with GFP) or TdTomato (TdT) as a control and analyzed puromycin immunostaining following DHPG stimulation. No significant differences were found in basal translation between uninfected and infected neurons. As expected, DHPG produced a significant increase in protein synthesis in both uninfected and TdT-infected neurons (Fig. 1C, black and red columns). In marked contrast, this increase in translation by mGluR1/5 stimulation was totally suppressed in Ras-DN-expressing neurons (Fig. 1C, purple columns). Taken together, these data indicate that Ras activity is essential for mGluR-dependent activation of both MAPK/Erk signaling and protein synthesis.

### Ras activity increases in spines during mGluR-LTD promoted by c-Src

After having established that Ras activity is required for mGluR-LTD expression, we aimed to evaluate whether mGluR-LTD induction indeed stimulates Ras activation in the hippocampus. To this end, we used two complementary approaches. First, Ras activation was determined using a commercial pull-down assay (Cytoskeleton #BK008) from hippocampal slice extracts after different treatments for plasticity induction. An increase in Ras activity was observed after mGluR1/5 activation with DHPG (Fig. 2A), to a similar extent as with a chemical LTP stimulation protocol (cLTP) (Otmakhov et al, 2004). In contrast, NMDAR-LTD induction with NMDA (Lee et al, 1998; Kamal et al, 1999) failed to engage Ras activation. Extracts were incubated with GDP or a non-hydrolysable form of GTP (GTPγS) as negative and positive controls, respectively.

Second, we wished to ascertain potential local activation of Ras during mGluR-LTD. To achieve this, we monitored Ras activation in dendritic spines using live imaging with a FRET system. We co-transfected Ras-GFP and the Ras activity FRET sensor FRas2-M (Oliveira and Yasuda, 2013), in hippocampal slice cultures using gene-gun technology. This FRET system is based on the Ras

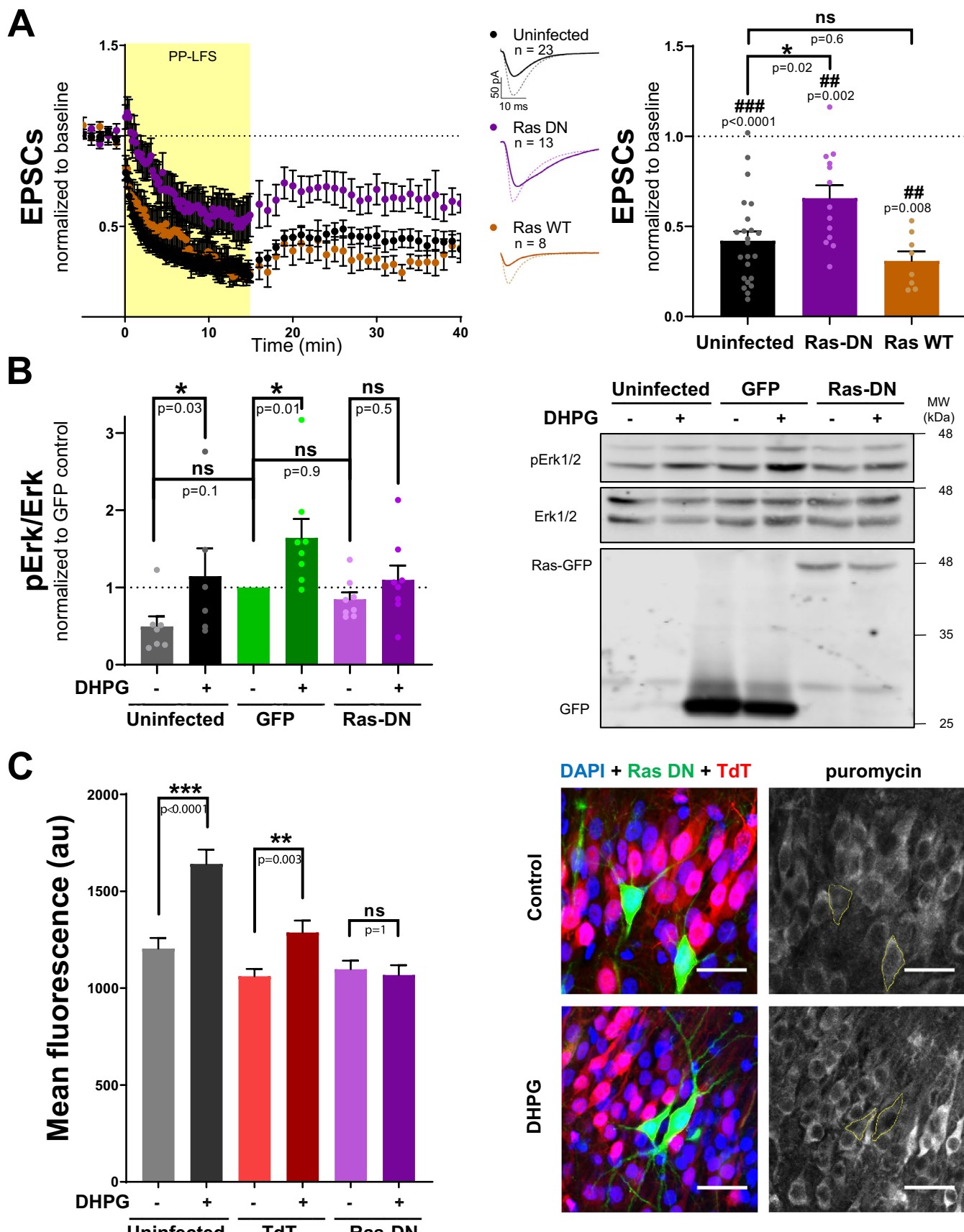

**Figure 1.  Effect of blocking Ras activity on mGluR-LTD and its related signaling.**

(A) Time course (left) of normalized AMPAR-mediated EPSCs during baseline and up to 40 min after mGluR-LTD induction (PP-LFS, 1 Hz) of CA1 hippocampal neurons from organotypic slices expressing Ras-DN (purple), Ras WT (orange) or control (uninfected) neurons. Representative traces are shown for baseline responses (dashed line) and for the last 10 min of the recording (thick line). Quantification (right) of the extent of depression 30–40 min after induction. Results are normalized to baseline and expressed as mean ± SEM; individual values are also represented. Wilcoxon signed-rank test (#) was used to statistically assess the extent of depression. Krustal-Wallis [H(2,44) = 10.0, p = 0.007] + Dunn's post-test (*) was used to evaluate significant differences between conditions. (B) Quantification (left) and representative Western blot (right) of phospho-Erk (pErk) and Erk signals from CA1 region of hippocampal slices uninfected (black) or infected with Sindbis viruses expressing GFP (green) or Ras-DN-GFP (purple) and stimulated with DHPG. Phospho-to-total ratio was calculated and normalized to GFP control condition. Results are expressed as mean ± SEM and individual values are also represented, n = 6–8 independent slice batches. Mixed effects analysis [F(1,7) = 19, p = 0.003] + Sidak's post-test (*) was used to evaluate DHPG effect for each condition. (C) Quantification (left) and representative images (right) of protein synthesis levels, measured as puromycin labeling from cultured hippocampal slices. Scale bars 30 µm. 80–156 neurons uninfected or infected with Sindbis viruses expressing TdTomato (TdT) or Ras-DN-GFP were analyzed. 10 control slices and 14 DHPG stimulated were used. Yellow lines circle Ras-DN-infected neurons. Results are represented as mean ± SEM. Mixed effects analysis [F(1,696) = 22.8, p < 0.0001] + Sidak's post-test (*) was used to evaluate DHPG effect for each condition. ns, non-significant. Source data are available online for this figure.

Binding Domain of Raf tagged with RFP, which binds only to active Ras thereby producing FRET. Ras activity, reported as FRET' signal (see Methods and Protocols), significantly increased in spines shortly after DHPG addition, and then gradually decreased after DHPG wash out (Fig. 2B). On the other hand, Ras activity was not significantly affected within the dendritic shaft. These results indicate that Ras is particularly activated in dendritic spines during mGluR-LTD. This effect is probably due to local Ras activation rather than Ras recruitment into spines, as total Ras-GFP signal in spines did not change during DHPG treatment or wash out (Fig. EV2).

We next wanted to determine the signaling mechanism mediating Ras activation downstream from mGluRs. We consider three main hypotheses: (i) Ras activation could be dependent on extracellular calcium and CaMKII (Ca²⁺/calmodulin-dependent protein kinase II) signaling, as is the case for LTP (Araki et al, 2015); (ii) it could be mediated through the canonical pathway, involving Gα_q protein, calcium release from the endoplasmic reticulum (ER) and PKC (protein kinase C) activation (Ribeiro et al, 2017); or (iii) it could rely on Src-mediated transactivation of tyrosine kinase receptors as they have been described to stimulate Ras activity upon G protein coupled receptor (GPCR) activation (Wang et al, 2007).

To address whether Ras activation depends on CaMKII, PKC or Src signaling, organotypic cultures were treated with different inhibitors before DHPG stimulation, and then Erk1/2 activation was evaluated as a downstream effector of Ras activation (since Erk1/2 phosphorylation during mGluR-LTD requires Ras activity; Fig. 1B). Blockade of CaMKII (with KN93) or PKC (with GF109203) did not affect Erk1/2 activation upon DHPG stimulation. In contrast, inhibition of Src family kinases (with PP2) suppressed DHPG-induced Erk1/2 activation (Fig. 2C). Furthermore, we also confirmed that PP2 completely blocks Ras activation induced by DHPG, using the Ras pull-down assay (Fig. 2D). Therefore, these results indicate that Src mediates Ras activation downstream from mGluR1/5.

Finally, electrophysiological recordings showed that PP2 also impaired mGluR-LTD in organotypic hippocampal slices (Fig. 2E) to a similar extent as Ras-DN (Fig. 1A). In addition, the c-Src selective inhibitor KBSRC4, produced a similar reduction in the extent of mGluR-LTD (Fig. 2E). Taken together, these results indicate that c-Src is required for mGluR-LTD, and is necessary for Ras activation and the subsequent MAPK/Erk signaling.

## Lack of H-Ras impairs object recognition, spatial and social memory

Our biochemical and electrophysiological data pointed to Ras as a key mediator of mGluR-LTD in the hippocampus. However, there is little information on potential specificity of Ras isoforms for synaptic or neuronal function, despite their known differences for tumor development or developmental disorders (Castellano and Santos, 2011). Although Ras isoforms are nearly ubiquitous, H-Ras is abundantly expressed in brain, muscle and skin (Leon et al, 1987; Castellano and Santos, 2011), and in fact is the most abundant isoform in the hippocampus (Manabe et al, 2000). Therefore, we started by evaluating brain functions of a knockout (KO) mouse for H-Ras previously generated by some of the authors of this study. These animals develop normally, reproduce at Mendelian ratios and show no major alterations in neuronal number or diversity (Esteban et al, 2001). As expected, hippocampal extracts from these mice showed a strong reduction in hippocampal total Ras levels (Fig. EV3A), with no differences in K-Ras expression, reinforcing the notion that H-Ras is the most abundant Ras isoform in the hippocampus. In addition, we found that the levels of different glutamate receptor subunits (mGluR1/5, GluA1/2, GluN1), along with a number of relevant signaling and scaffold synaptic proteins (Homer-1, SynGAP, Erk1/2, Akt or Src), were no different between H-Ras KO mice and their WT littermates. We then proceeded to evaluate the role of H-Ras in animal behavior and cognition, using different behavioral tasks.

First, novelty-induced locomotor activity was evaluated in the open field test. H-Ras KO mice locomotor activity was comparable to control mice and habituated normally to the arena over time (Fig. 3A). Thus, no differences between genotypes were found in the distance traveled (Figs. 3A and EV3B), the mean speed (Fig. EV3C), or the time spent in the different areas of the arena (Fig. 3B). Working memory, as evaluated by spontaneous alternations in the Y-maze, was not significantly different between H-Ras-deficient and WT mice (Fig. 3C).

Episodic memory was tested using an object recognition test. After an initial presentation phase ("familiarization"), animals are exposed to the same objects 1 h later ("reexposure"). We observed that while control animals spent significantly less time exploring the objects in the reexposure phase, as compared to the familiarization phase, H-Ras KO mice virtually spent the same time exploring the objects in both phases of the test (Fig. 3D, left

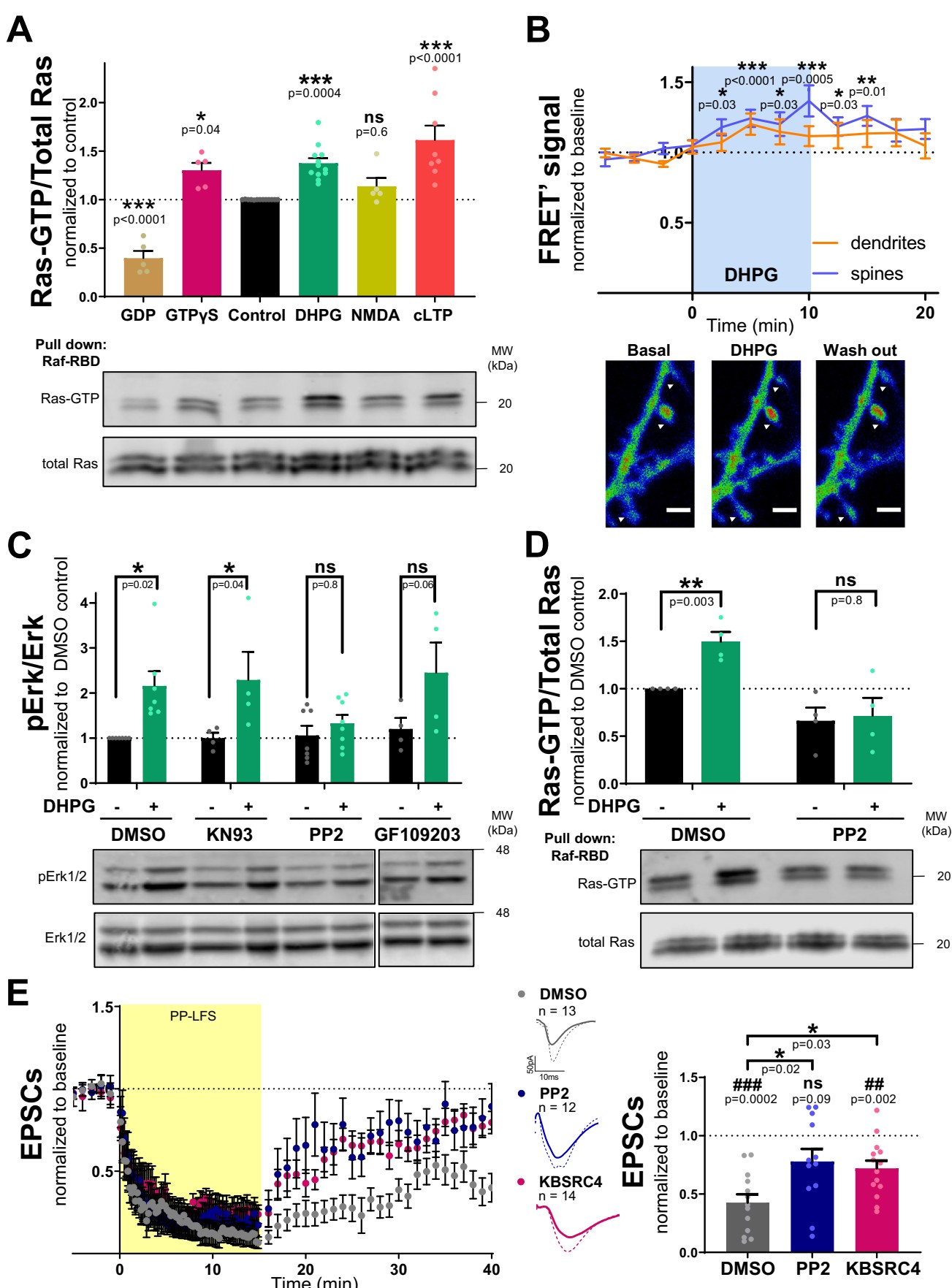

**Figure 2.   Ras activation during mGluR-LTD.**

(A) Quantification (upper panel) and representative Western blot (lower panel) of active Ras pulled-down vs total Ras from hippocampal slice cultures after inducing different chemical plasticity protocols. Both Ras bands were quantified together and results were normalized to control, non-stimulated slices (dashed line) and expressed as mean ± SEM, $n = 5$–13 independent slice batches; individual values are also represented. One-way ANOVA [$F_{(5,42)} = 23.7$, $p < 0.0001$] + Dunnett's post-test (*) was used to evaluate Ras activation compared to control. (B) Time course of Ras activity upon DHPG stimulation in spines (blue, $n = 50$) and dendrites (orange, $n = 15$) of CA1 pyramidal cells ($n = 10$) evaluated by FRET. Results are represented as mean ± SEM. Representative images of the FRET signal from dendritic branches of CA1 neurons coexpressing H-Ras-GFP and FRas2-M obtained during the time course are shown. Scale bars 2 μm. White arrows point dendritic spines. Kruskal–Wallis [$H_{(11,571)} = 53.8$, $p < 0.0001$ for spines, and $H_{(11,178)} = 14.1$, $p = 0.2$ for dendrites] + Dunn's post-test (*) was used to evaluate differences across time (comparing to baseline). (C) Quantification (upper panel) and representative Western blot (lower panel) of phospho-Erk (pErk) and Erk signals from hippocampal slices treated with vehicle (DMSO) or inhibitors of CaMKII (KN93), Src family (PP2) or PKC (GF109203) in response to DHPG stimulation (green columns). The phospho-to-total ratio was calculated and normalized to DMSO control condition. Results are expressed as mean ± SEM and individual values are also represented, $n = 4$–8 independent slice batches. Mixed effects analysis [$F_{(1,7)} = 30.0$, $p = 0.0009$] + Sidak's post-test (*) was used to evaluate DHPG effect for each condition. (D) Quantification (upper panel) and representative Western blot (lower panel) of active Ras pulled-down vs total Ras from hippocampal slice cultures treated with vehicle DMSO or PP2 and stimulated with DHPG. Results are normalized to vehicle non-stimulated slices and expressed as mean ± SEM, $n = 4$ independent slice batches; individual values are also represented. Mixed effects analysis [$F_{(1,6)} = 18.6$, $p = 0.005$] + Sidak's post-test (*) was used to evaluate DHPG effect for each condition. (E) Time course (left) of normalized AMPAR-mediated EPSCs from baseline to 40 min after mGluR-LTD induction (PP-LFS, 1 Hz) from organotypic slices treated with DMSO (gray), Src family inhibitor PP2 (blue) or c-Src inhibitor KBSRC4 (pink). Representative traces are shown for baseline responses (dashed line) and for the last 10 min of the recording (thick line). Quantification (right) of the extent of depression 30–40 min after induction. Results are normalized to baseline and expressed as mean ± SEM; individual values are also represented. Wilcoxon signed-rank test (#) was used to assess statistically the extent of depression. Kruskal–Wallis [$H_{(2,39)} = 8.2$, $p = 0.02$] + Dunn's post-test (*) was used to evaluate significant differences between conditions. ns, non-significant. Source data are available online for this figure.

panel). This result argues for a critical role of H-Ras in object familiarization. Interestingly, this cognitive function has been associated with mGluR-LTD (Zhu et al, 2018). In the last phase of the test, a new object is introduced, replacing one of the previous objects. In this phase, both WT and KO animals successfully discriminated the novel object from the familiar one (Fig. 3D, right panel).

In addition, we evaluated contextual fear conditioning to test spatial memory, in which the hippocampus has a prominent role. Interestingly, we found a significant impairment of contextual fear memory 24 h after training in H-Ras KO mice (Fig. 3E, left panel), despite having normal freezing responses during training (Fig. EV3D). In contrast, no differences in freezing time between WT and KO mice were observed after cued fear conditioning (Fig. 3E, right panel), a task that mostly relies on amygdala functioning, but not on hippocampal activity (Phillips and LeDoux, 1992; Oh and Han, 2020). This result strongly suggests that H-Ras is specifically required for the spatial features of fear memory formation.

Lastly, sociability and social memory were evaluated by the three-chambered social approach test (Fig. 3F). In the social preference task, both control and KO mice preferentially explored an unfamiliar mouse instead of an inanimate object (Fig. 3F, left panel). However, H-Ras KO mice failed to discriminate between a familiar and a novel mouse, as control mice did (Fig. 3F, right panel). Therefore, H-Ras is also required for social memory.

## H-Ras is required postsynaptically for mGluR-LTD

The development of the H-Ras KO mouse and its cognitive deficits as demonstrated in our studies opened the possibility of testing the role of this isoform in synaptic function. There is still a dearth of information on isoform-specific functions of Ras in the brain. To address this, we performed field recordings of CA3-to-CA1 synaptic responses in acute hippocampal slices from young adult WT and H-Ras KO mice. Input/output curves of field excitatory postsynaptic potentials (fEPSP) showed a small but statistically significant decrease in basal synaptic transmission in H-Ras KO

mice compared to WT mice (Fig. 4A). However, no differences were found in presynaptic fiber volley amplitude (Fig. EV4A) or in paired-pulse ratio (Fig. 4B), suggesting normal presynaptic function in the absence of H-Ras.

We next evaluated the contribution of H-Ras to mGluR-LTD. First, we found that DHPG-induced Ras activation was completely suppressed in H-Ras KO mice (Fig. EV4B), which confirms that it is the only Ras isoform activated by mGluR stimulation. In addition, mice lacking H-Ras showed a clear impairment in mGluR-LTD induced by PP-LFS (Fig. 4C). This is consistent with our results with Ras-DN overexpression (Fig. 1A). Furthermore, these results establish that H-Ras is uniquely involved in this form of plasticity, since its contribution cannot be compensated by the other Ras isoforms present in the brain. Nevertheless, we should point out that these H-Ras-deficient mice are deprived of H-Ras in all cell types and throughout development. Therefore, in order to test the direct role of H-Ras in mGluR-LTD, we attempted a rescue experiment. To this end, we re-expressed H-Ras WT in postsynaptic CA1 neurons of organotypic hippocampal slice cultures from H-Ras KO mice. As shown in Fig. 4D, H-Ras WT re-expression (orange symbols) restored mGluR-LTD over the depression levels obtained with uninfected KO neurons (blue symbols). In fact, the extent of depression was similar to that of uninfected neurons from WT organotypic slices (Fig. 1A). On the one hand, this result confirms that H-Ras is directly required for mGluR-LTD. On the other hand, this experiment indicates that H-Ras requirement for mGluR-LTD relies on its postsynaptic localization, since H-Ras was re-introduced exclusively in the postsynaptic CA1 neuron, within a full H-Ras KO background. This is a relevant point, considering that mGluR-LTD is based on both presynaptic and postsynaptic components (Nosyreva and Huber, 2005; Sánchez-Castillo et al, 2022), which may explain why mGluR-LTD was not fully ablated in HRas KO mice.

Palmitoylation of H-Ras at the C-terminus is essential for membrane localization to endocytic compartments (Gomez and Daniotti, 2005), and complete abolition of palmitoylation results in H-Ras retention at the ER and Golgi (Pedro et al, 2017). In order to test the relevance of H-Ras subcellular localization for mGluR-LTD, we

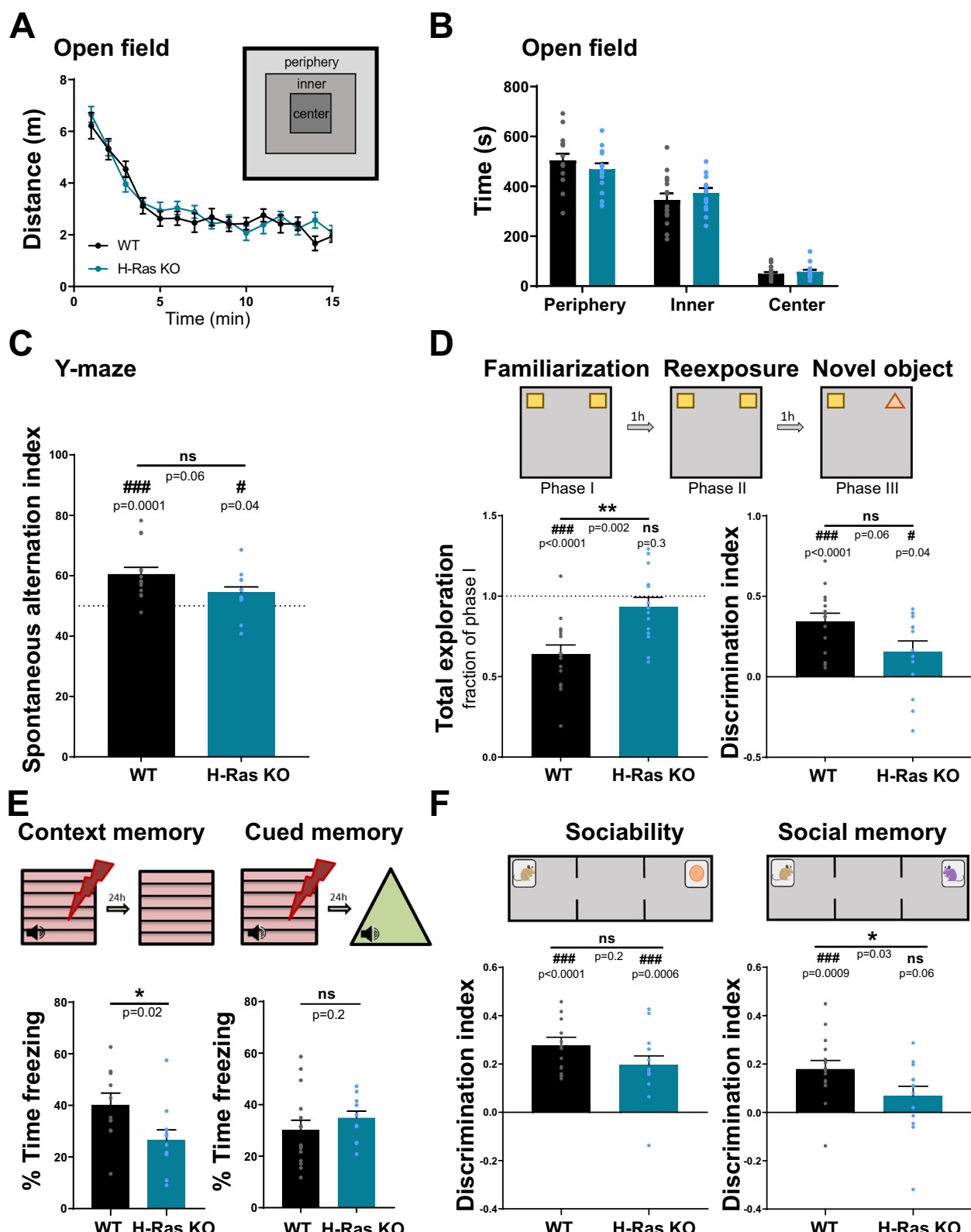

repeated the rescue experiment with a recombinant H-Ras in which its two C-terminal palmitoylation sites have been mutated (C181S, C184S). As shown in Fig. 4D (green symbols), expression of H-Ras C181,184S in postsynaptic CA1 neurons failed to rescue mGluR-LTD

in H-Ras KO slices, with an extent of depression similar to uninfected neurons. This result indicates that H-Ras membrane localization is critical for its role in this form of plasticity. In fact, confocal images of CA1 neurons expressing recombinant GFP-H-Ras indicate that H-Ras

**Figure 3. H-Ras KO mice have altered object recognition, spatial and social memory.**

(A) Distance traveled in the open field test, divided in 1 min intervals and diagram of the open field arena (upper inset). (B) Total time spent in the different open field areas. (C) Spontaneous alternation index in the Y-maze test. (D) Quantification of time sniffing the objects during the reexposure phase, compared to the familiarization phase (left) and discrimination index for the novel object in the third phase (right) (diagram above the graphs). (E) Contextual fear conditioning test diagram (upper left) and percentage of time spent freezing upon reexposure to the context 24 h after training (lower left). Cued fear conditioning test diagram (upper right) and percentage of time spent freezing upon reexposure to the shock-paired tone 24 h after training (lower right). (F) Social preference task diagram (upper left) and discrimination index of the subject (lower left). Social novelty preference diagram (upper right) and discrimination index of the novel subject (lower right). Mixed effects analysis was used for (A, B) [F(1,27) = 0.2, p = 0.7 for (A) and F(1,28) = 1.3, p = 0.3 for (B)]. For (C–F), Wilcoxon signed-rank test (#) was used to assess statistically the pertinent learning index (spontaneous alternation, reduction in exploration or discrimination indexes). Mann-Whitney test (*) was used to evaluate differences between genotypes; ns, non-significant. (A–D, F): n = 15 mice for WT and 14 mice for H-Ras KO. (E) Context memory: n = 10 mice for WT and 11 mice for H-Ras KO. (E) Cued memory: n = 15 mice for WT and 11 mice for H-Ras KO. Source data are available online for this figure.

C181,184S localization in dendritic spines is significantly reduced, as compared to H-Ras WT (Fig. 4E), despite both recombinant proteins being expressed at similar levels in hippocampal homogenates (Fig. EV4C).

## K-Ras, not H-Ras, is the isoform required for LTP

As previously mentioned, all Ras isoforms are membrane proteins, which are irreversibly farnesylated at the C-terminus, but differ in their carboxy-terminal hypervariable region (Fig. 4F), which undergoes further posttranslational modifications essential for Ras intracellular trafficking (Prior and Hancock, 2001). The fact that H-Ras palmitoylation is required for its function during mGluR-LTD (Fig. 4D) suggests that Ras subcellular localization may provide functional specificity for different isoforms in synaptic function (and explain why other Ras isoforms cannot compensate for H-Ras during mGluR-LTD). We then wondered whether H- and K-Ras (the two most abundant Ras isoforms in the brain (Newlaczyl et al, 2017)) had different subcellular and synaptic distribution in hippocampal neurons. To address this question, we performed subcellular fractionation in hippocampal extracts from adult mice (Fig. 5A). K-Ras levels were determined using a specific K-Ras antibody, whereas H-Ras levels were assessed from the comparison between WT and H-Ras KO mice using a pan-Ras antibody (since there was no specific H-Ras antibody commercially available, as tested with H-Ras KO extracts). Interestingly, H-Ras was highly concentrated in the non-postsynaptic density (non-PSD) fraction of synaptosomes, and almost absent from microsome and PSD fractions, a result consistent with the literature (Manabe et al, 2000). In contrast, K-Ras was broadly distributed across the three membrane compartments, with clear presence at the PSD fraction. Interestingly, while H-Ras subcellular distribution was not generally affected by DHPG treatment, K-Ras concentration at the PSD was significantly reduced in response to mGluR stimulation (Fig. EV5A). These important differences in synaptic distribution for the two main Ras isoforms in the brain support the notion that they may fulfill distinct synaptic functions.

Perhaps one of the most established synaptic functions of Ras is to support Erk activation during LTP (Zhang et al, 2018; Zhu et al, 2002; Araki et al, 2015). In most cases, this role has been assumed to be carried out by H-Ras, based on overexpression of recombinant proteins, or simply because H-Ras is abundantly expressed in the brain. Therefore, we decided to directly test this hypothesis with the H-Ras KO mice. As shown in Fig. 5B, NMDAR-LTP induced by theta-burst stimulation (TBS) in acute

hippocampal slices from H-Ras KO mice was completely preserved, and similar to WT control littermates. Although unexpected, this result is consistent with the absence of an LTP impairment previously described with an independent H-Ras KO mouse (Manabe et al, 2000). Therefore, in conjunction with multiple studies providing evidence on the role of Ras signaling for LTP, these results strongly suggest that a different Ras isoform must be responsible for NMDAR-dependent LTP (at least at CA3-to-CA1 hippocampal synapses).

Based on the differing subcellular fractionation between K-Ras and H-Ras (and the specific presence of K-Ras at the PSD), we hypothesized that K-Ras might be the isoform required for LTP. To address this possibility, we conducted bilateral injections of adeno-associated viruses (AAVs) expressing Cre recombinase under the CaMKIIα promoter (AAV-CaMKII-mCherry-Cre) into the hippocampus of adult K-Ras$^{lox/lox}$ mice. These floxed mice were used to bypass the embryonic lethality of the complete K-Ras KO (Koera et al, 1997; Johnson et al, 1997). Four weeks post-injection, mCherry-Cre expression was observed throughout the pyramidal cell layers in the hippocampus (Fig. EV5B), accompanied by a nearly 50% reduction in K-Ras protein levels (Fig. EV5C). To note, KO efficiency is not expected to be complete, as K-Ras was only ablated from neurons (K-Ras nKO), and hence should still be present in glial cells and cerebral vasculature. Expression levels of total Ras were not significantly decreased, as compared to saline-injected (control) mice, consistent with the notion that H-Ras is the most abundant Ras isoform in the hippocampus. We then performed field recordings from acute hippocampal slices of these mice to examine the functional consequences of the neuronal removal of K-Ras. While no differences were found in basal transmission (Fig. EV5D), K-Ras nKO showed a strong impairment of LTP compared to control saline-injected mice (Fig. 5C). Consistent with this result, K-Ras activity was significantly enhanced by cLTP stimulation, as assessed by pull-down assay (Fig. EV5E). Interestingly, mGluR-LTD remained unaffected in K-Ras nKO mice (Fig. 5D). Taken together, these results support a clear functional specialization of neuronal Ras isoforms for the control of synaptic plasticity, with K-Ras being required to sustain LTP and H-Ras mediating mGluR-LTD.

## Discussion

In this study, we report a previously overlooked functional specialization of neuronal Ras isoforms in synaptic plasticity. Using genetically modified animals, we show that H-Ras

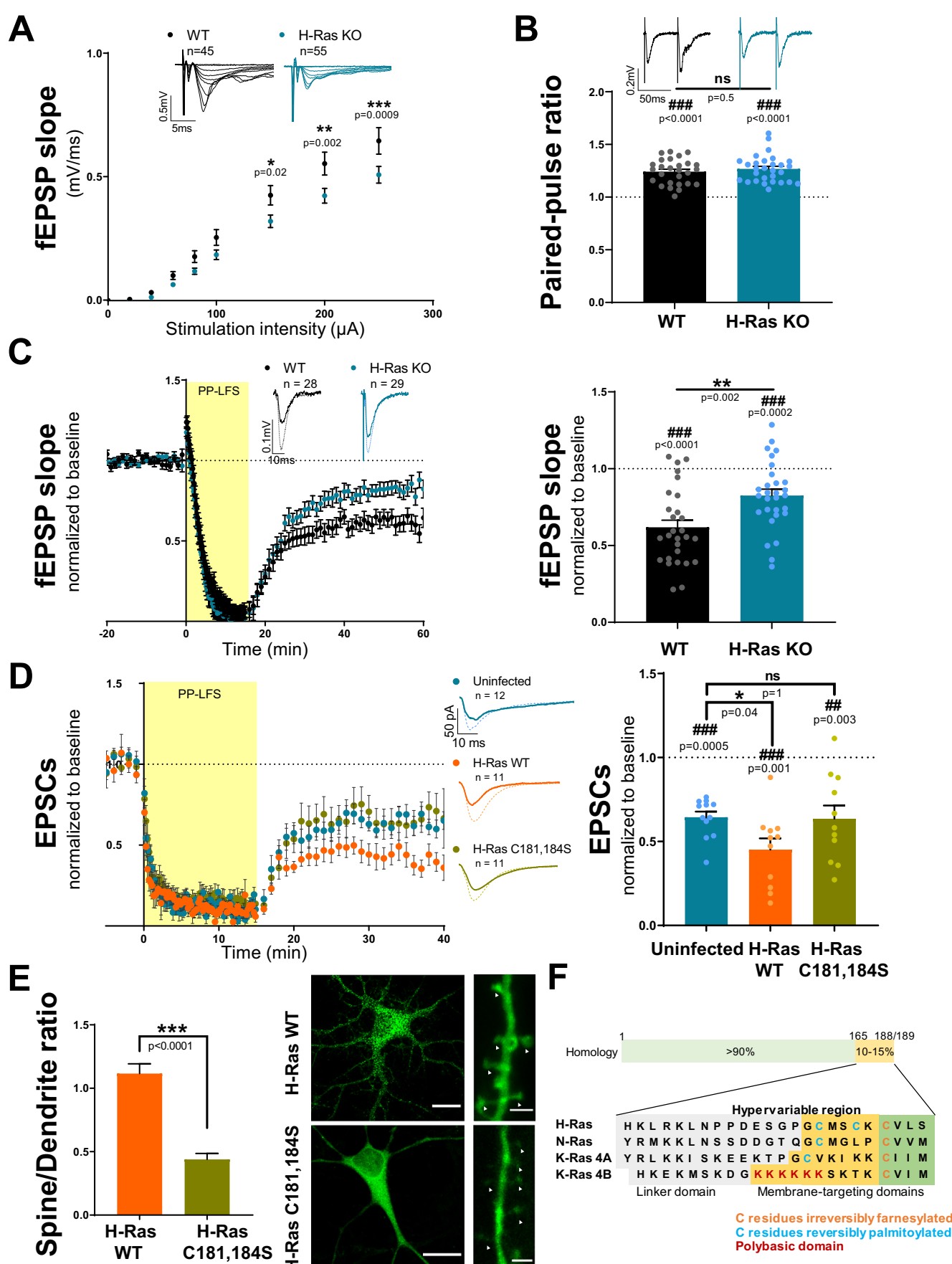

**Figure 4.  H-Ras is required postsynaptically for mGluR-LTD.**

(A) Input/output curves of fEPSP slopes vs stimulation intensities from acute hippocampal slices performed in CA1 area. Representative traces are shown in the upper part. Mean ± SEM; Mixed effects analysis [$F_{(1,98)}$ = 6.0, $p$ = 0.02] + Sidak's post-test (*) was used to evaluate differences between genotypes. (B) Quantification of the paired-pulse ratio of fEPSPs. Mean ± SEM and individual values are represented. Representative traces are shown in the upper part. Wilcoxon signed-rank test (#) was used to assess statistically the degree of facilitation. Mann–Whitney test (*) was used to evaluate significant differences between genotypes. $n$ = 27 (WT) and 29 (H-Ras KO) slices. (C) Time course (left) of normalized fEPSPs from baseline to 60 min after mGluR-LTD induction (PP-LFS, 1 Hz). Representative traces of baseline responses (dashed line) and for the last 10 min of the recording (thick line) are shown as inset. Quantification (right) of the extent of depression 50–60 min after induction. Results are normalized to baseline and expressed as mean ± SEM; individual values are also represented. Wilcoxon signed-rank test (#) was used to assess statistically the depression. Mann–Whitney test (*) was used to evaluate statistical differences between genotypes. (D) Whole-cell voltage-clamp recordings of CA1 pyramidal neurons from hippocampal organotypic slices of H-Ras KO animals. Recordings were carried out from uninfected neurons (blue) and neighboring neurons infected with Sindbis virus expressing Ras WT (orange) or Ras C181,184S (green). Time course (left) of normalized AMPAR-mediated EPSCs from baseline to 40 min after mGluR-LTD induction (PP-LFS, 1 Hz). Representative traces of baseline responses (dashed line) and for the last 10 min of the recording (thick line) are shown as inset. Quantification (right) of the extent of depression 30–40 min after induction. Results are normalized to baseline and expressed as mean ± SEM; individual values are also represented. Wilcoxon signed-rank test (#) was used to assess statistically the extent of depression. Kruskal–Wallis [$H_{(2,30)}$ = 7.2, $p$ = 0.03] + Dunn's post-test (*) was used to evaluate significant differences between conditions. (E) Quantification of GFP-Ras spine/dendrite ratio (left) $n$ = 25 (H-Ras) and 32 (H-Ras C181,184S) spines. Results are expressed as mean ± SEM. Representative images (right) of CA1 pyramidal neurons and its dendritic branches expressing GFP-Ras WT and GFP-Ras C181,184S. Scale bars 15 (left) and 2 (right) μm. White arrows point dendritic spines. Mann–Whitney test (*) was used to assess statistical differences. ns, non-significant. (F) Comparative scheme of the protein sequence at the C-terminal hypervariable region for the different Ras isoforms, including amino acid residues that undergo post-translational modifications such as farnesylation and palmitoylation. Modified from (Prior and Hancock, 2001). Source data are available online for this figure.

contributes to mGluR-LTD (and is dispensable for NMDAR-LTP), while K-Ras is required for NMDAR-LTP (and does not participate in mGluR-LTD). Interestingly, this functional specificity correlates with a marked difference in the subcellular distribution of these isoforms, with H-Ras being strongly enriched in non-PSD membranes of synaptosomes, and K-Ras being more evenly distributed, including in the PSD. These results are also consistent with the selective activation of H-Ras and K-Ras following chemical induction of mGluR-LTD and LTP, respectively. Also, in terms of signaling, we found that H-Ras functions downstream of c-Src activity and mediates postsynaptic Erk activation and protein synthesis during mGluR-LTD (see model in Fig. 6).

NMDAR-LTP and mGluR-LTD may initially be considered antagonistic processes as they promote opposing effects on synaptic efficiency. Yet, both share critical signaling pathways such as Akt/mTOR, MAPK/Erk activation and induction of protein synthesis, even though these result in opposite outcomes for AMPAR membrane expression and synaptic transmission (Malenka and Bear, 2004; Reiner and Levitz, 2018; Ye and Carew, 2010). In order to understand how shared signaling pathways orchestrate contrary effects on synaptic plasticity, we ought to dissect the specific mechanisms mediated by their molecular players.

In the present study, we report that H-Ras acts postsynaptically to support mGluR-LTD. The synaptic locus for H-Ras action is relevant, because one could have assumed that PI3K activity and Ras signaling would be connected for mGluR-LTD, since PI3K is also required for this form of plasticity (Hou and Klann, 2004) and Ras is a well-known upstream effector of PI3K (Kodaki et al, 1994; Rodriguez-Viciana et al, 1994). However, we recently found that PI3K is involved presynaptically for mGluR-LTD, by controlling neurotransmitter release (Sánchez-Castillo et al, 2022). Furthermore, this role is carried out by the PI3K isoform p110β (Sánchez-Castillo et al, 2022), which is not effectively activated by Ras (Fritsch et al, 2013). Therefore, our results indicate that both PI3K and H-Ras are required for mGluR-LTD, yet they act on different synaptic compartments (postsynaptic for Ras and presynaptic for PI3K), and therefore their signaling for this form of plasticity is not connected. Indeed, this model is consistent with previous work showing the coexistence of pre- and postsynaptic mechanisms in

mGluR-LTD (Nosyreva and Huber, 2005; Sánchez-Castillo et al, 2022) and may explain the observed residual depression when Ras activity/signaling was blocked. Instead, our data indicate that the postsynaptic contribution of H-Ras to mGluR-LTD relies on the activation of Erk signaling and protein synthesis, which are known to be required for mGluR-LTD expression (Banko et al, 2006; Sanderson et al, 2016; Huber et al, 2000; Gallagher et al, 2004). Furthermore, we report that Ras activation upon mGluR-LTD induction is mediated by c-Src activity. Thus, our data allows us to propose a signaling relay of mGluR/c-Src/H-Ras/MAPK/Erk that transduces mGluR activation into mRNA translation and AMPAR internalization, possibly via Arc synthesis (Waung et al, 2008).

In terms of molecular mechanisms, it is also relevant that H-Ras dynamically associates with recycling endosomes in a Rab5- and Rab11-dependent manner (Roy et al, 2002; Gomez and Daniotti, 2005). Group I mGluRs show rapid internalization upon ligand exposure, entering the endosome recycling compartment and trafficking back to the plasma membrane (Mahato et al, 2015, 2018; Pandey et al, 2014). Hence, it is reasonable to envision that mGluR-Ras signaling for mGluR-LTD occurs on endosomal compartments. This interpretation is consistent with our observation that the palmitoylation of H-Ras is required for mGluR-LTD, because this lipid modification is known to be essential for proper regulation of H-Ras intracellular trafficking (Agudo-Ibáñez et al, 2015; Pedro et al, 2017). In this regard, it has been shown that β-arrestin2 mediates mGluR signaling (Stoppel et al, 2017) and is required for mGluR-LTD in CA1 neurons (Eng et al, 2016). Although further research is required to confirm it, we hypothesize that β-arrestin2 recruits and promotes Src activation upon mGluR activation in CA1, as it has been shown in CA3 (Eng et al, 2016) and for other ligand-activated GPCRs (i.e., M2 muscarinic receptors, D2 dopamine receptors or β2 adrenergic receptors) (Kim et al, 2001; Luttrell et al, 1999, 2001; Pakharukova et al, 2020). Subsequently, Src would activate H-Ras (Guo et al, 2006; Luttrell et al, 1996; Peavy et al, 2001; Wang et al, 2018, 2007), thereby promoting Raf1/MEK/Erk signaling cascade. Interestingly, it has been shown that H-Ras decreases Src activity in brain slices (Thornton et al, 2003), which may serve as a negative feedback to control this pathway after stimulation.

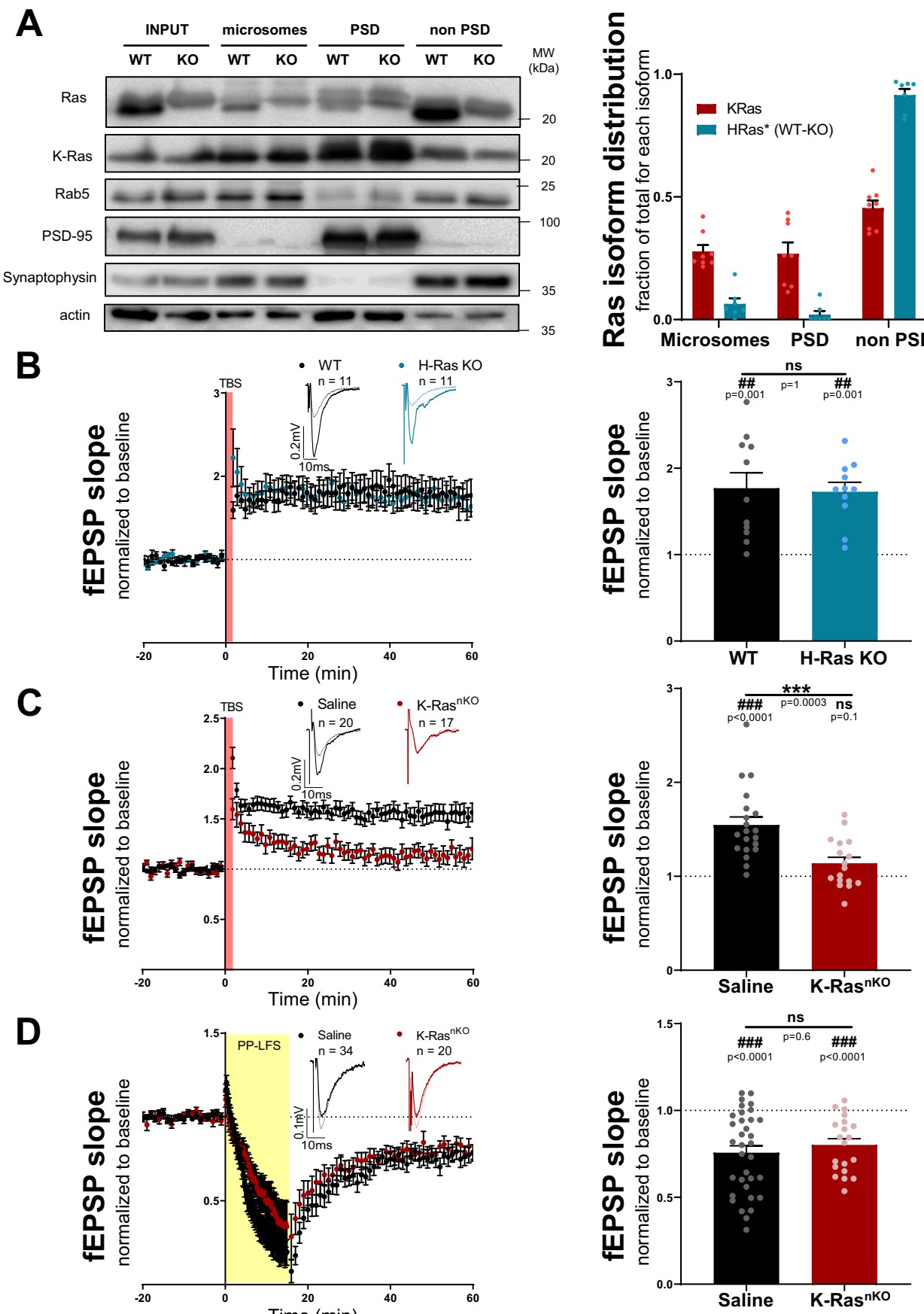

**Figure 5. K-Ras, not H-Ras, is required for NMDAR-LTP.**

(A) Representative Western blot (left) of hippocampal total Ras, K-Ras, Rab5, PSD-95 and synaptophysin subcellular fractionation. H- and K-Ras distribution in microsomes, PSD and non-PSD fractions (right) were quantified as percentage of total protein (from inputs) for each isoform. H-Ras signal was obtained by subtracting total Ras signal of H-Ras KO mice from that of WT mice. Results expressed as mean ± SEM, $n = 7$–8 animals per condition; individual values are also represented. (B–D) Time course (left) of normalized fEPSPs from baseline to 60 min after NMDAR-LTP induction (TBS) (B, C) or mGluR-LTD induction (PP-LFS, 1 Hz) (D), recorded from H-Ras KO mice (B) or K-Ras neuronal KO mice (C, D). Representative traces of baseline responses (dashed line) and for the last 10 min of the recording (thick line) are shown as insets. Quantification (right) of the extent of potentiation/depression 50–60 min after induction. Results are normalized to baseline and expressed as mean ± SEM; individual values are also represented. Wilcoxon signed-rank test (#) was used to assess statistically the potentiation/depression. Mann-Whitney test (*) was used to evaluate statistical differences between genotypes. ns, non-significant. Source data are available online for this figure.

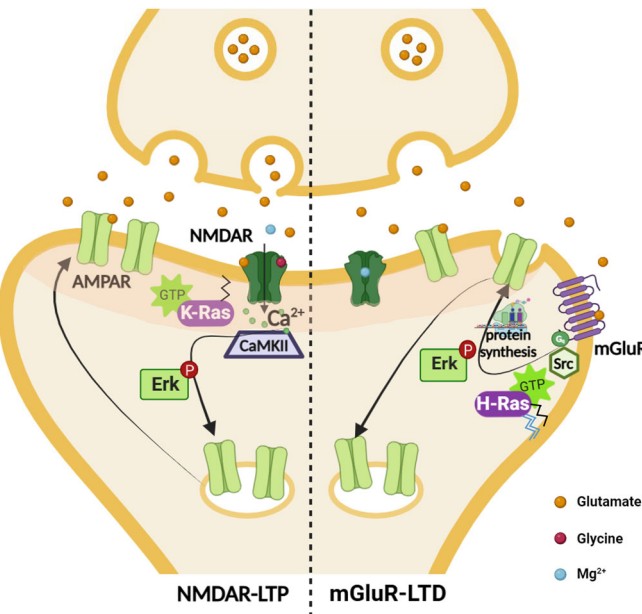

**Figure 6. Schematic model for Ras signaling and isoform specificity during NMDAR-LTP and mGluR-LTD.**

NMDAR-LTP is mediated by K-Ras, coupling Ca²⁺ and CaMKII activity to Erk signaling and AMPAR delivery into synapses (Araki et al, 2015; Chen et al, 1998; Qin et al, 2005). Conversely, mGluR-LTD involves Src-mediated H-Ras activation and subsequent stimulation of Erk signaling and protein synthesis leading to removal of synaptic AMPAR. Created with Biorender.

The intracellular compartmentalization of synaptic signaling is also compatible with the Ras isoform specialization we have found for LTD versus LTP. In the case of NMDAR-LTP, it has been previously shown that Ras is activated postsynaptically, as a consequence of the phosphorylation of SynGAP by CaMKII (Chen et al, 1998) and its dispersion away from the PSD (Araki et al, 2015), removing the local inhibition of Ras activity at the synapse (but see also (Araki et al, 2024)). Subsequently, active Ras promotes Erk and PI3K signaling (Qin et al, 2005). Incidentally, the direct coupling between Ras and PI3K signaling is indeed feasible in the case of LTP, because we have previously described that PI3K activity is required for LTP postsynaptically and is mediated by the p110α isoform (Sánchez-Castillo et al, 2022), which is readily activated by Ras (Fritsch et al, 2013). Nevertheless, and contrary to most expectations, we have found that the Ras isoform required for LTP is K-Ras (and not H-Ras). K-Ras (particularly K-Ras4B, the splice variant present in the brain) has a very different membrane targeting from H-Ras, based on its polybasic hypervariable region

(Prior and Hancock, 2001). In fact, we have found that K-Ras is present at the PSD fraction of hippocampal extracts, in marked contrast with H-Ras (see also (Manabe et al, 2000)). Furthermore, K-Ras has been shown to translocate in and out of the plasma membrane in hippocampal neurons in an activity-dependent manner, via the interaction between its polybasic motif and Ca²⁺/Calmodulin (Fivaz and Meyer, 2005). In addition, Ca²⁺/Calmodulin mediates the association of K-Ras with recycling endosomes engaged in AMPAR synaptic delivery during LTP (Loo et al, 2014). To note, by using recombinant Ras mutants targeted to different membrane compartments, it has been proposed that Ras promotes MAPK/Erk signaling from lipid rafts during LTP (Zhang et al, 2018). However, we now know that LTP is mediated by K-Ras, which endogenously does not reach lipid rafts (Prior and Hancock, 2001). Also, disruption of lipid rafts interferes with H-Ras, but not K-Ras, signaling (Roy et al, 1999). As a potential explanation for this discrepancy, we should consider that the PSD (a detergent-resistant membrane fraction) contains raft and non-raft domains (Hering et al, 2003). Therefore, it is possible that targeting recombinant Ras to lipid rafts may also result in partial targeting to the PSD, where endogenous K-Ras is present, and interfere with its function. Indeed, this may also explain why Ras-DN based on H-Ras sequence impaired LTP in previous studies (Zhu et al, 2002; Zhang et al, 2018). Ras-DN has increased affinity for GDP (Feig and Cooper, 1988; Nassar et al, 2010), and hence it may sequester Ras activators (i.e., Ras-GEFs) away from endogenous Ras and cause unspecific side effects on other Ras isoforms, including K-Ras (Matallanas et al, 2003).

The specialization of K-Ras for LTP is also supported by its molecular signaling. Thus, among the Raf isoforms transducing Ras signaling towards the MAPK/Erk pathway (A-Raf, B-Raf and C- or Raf-1), B-Raf is preferentially coupled to K-Ras versus H-Ras (Terrell et al, 2019). Consistently, B-Raf is required for hippocampal LTP, learning and memory (Chen et al, 2006). Our results are also in line with a pharmacogenetic approach showing that K-Ras-deficient mice are more sensitive to subthreshold concentrations of MEK inhibitors in terms of LTP expression and spatial memory (Ohno et al, 2001). In addition, there are several Ras-GEFs that have been detected in the PSD, including RasGRF1 (Sturani et al, 1997) and Sos1 (Suzuki et al, 1999). However, previous work has demonstrated a specific role for Ras-GRF2 in LTP, while RasGRF1 has been related to NMDA-dependent LTD (Li et al, 2006). Conversely, which Ras-GEFs are involved in mGluR-LTD still remains to be addressed. One potential candidate is Ras-GRF1, which activates H-Ras but not N- or K-Ras (Matallanas et al, 2003), and moreover it is activated by c-Src (Kiyono et al, 2000), thereby mechanistically linking both proteins. Another one is Sos-1, which can activate all Ras isoforms but more efficiently in the case of

H-Ras (Jaumot et al, 2002). In any event, it is likely that the functional specialization between H-Ras and K-Ras is driven by their distinct subcellular distribution, which may bias their association with specific binding partners.

Finally, our results on Ras isoforms and synaptic plasticity also bear implications for cognitive function and neurodevelopmental disorders. This is especially relevant for RASopathies, particularly those caused by activating mutations in H-Ras (Costello syndrome) and K-Ras (Noonan and cardiofaciocutaneous syndromes), which are the most prevalent group of neurodevelopmental disorders (Kim and Baek, 2019; Rauen, 2013; Hebron et al, 2022). We have found that H-Ras-deficient mice have impaired object recognition, along with deficits in spatial learning and social memory. Interestingly, a mouse model for Costello syndrome expressing a constitutively active form of H-Ras (G12V) presented deficits in spatial learning, in addition to impaired mGluR-LTD (and normal LTP) (Schreiber et al, 2017). Therefore, it appears that either up- or downregulation of H-Ras signaling impairs mGluR-LTD and causes cognitive deficits. Along the same lines, mutant mice with overactive K-Ras recapitulate the Noonan syndrome (Hernández-Porras et al, 2014), and display impaired LTP and deficits in spatial memory (Papale et al, 2017; Ryu et al, 2020). Therefore, taken together, these findings support a model in which Ras isoforms serve as molecular switches that provide functional specification for synaptic plasticity (LTP versus LTD), with relevant implications for cognitive function and neurodevelopmental disorders.

# Methods

**Reagents and tools table**

| Reagent/Resource | Reference or Source | Identifier or Catalog Number |
|---|---|---|
| **Experimental models** | | |
| Wistar rats | Charles River | Wistar Han IGS |
| H-Ras KO mice | Eugenio Santos Lab | Hras$^{tm1Esn}$, MGI:2385945 |
| K-Ras$^{lox/lox}$ | Mariano Barbacid Lab | |
| **Recombinant DNA** | | |
| pEGFP-HRas WT | Mariano Barbacid Lab | |
| pEGFP-HRas DN (S17N) | This study | |
| pEGFP-HRas C181,184S | This study | |
| FRas2-M (FRET probe) | Addgene | #45149 |
| AAV5-CaMKII-Cre-mCherry | Gene Therapy Vector Core, University of North Carolina | serotype 5 |
| **Antibodies** | | |
| Ras | Cell signaling | #3965 |
| pErk1/2(T202/Y204) | Cell signaling | #9106 |
| Erk1/2 | Cell signaling | #4695 |
| pAkt (T308) | Cell signaling | #2965 |
| pAkt (S473) | Cell signaling | #4060 |
| Akt | Cell signaling | #2920 |

| Reagent/Resource | Reference or Source | Identifier or Catalog Number |
|---|---|---|
| S6 | Cell signaling | #2217 |
| Src | Cell signaling | #2108 |
| SynGAP | Cell signaling | #3200 |
| K-Ras | Sigma/Merck-Millipore | WH0003845M1 |
| mGluR5 | Sigma/Merck-Millipore | AB5675 |
| mGluR1 | Sigma/Merck-Millipore | #07-617 |
| GluN1 | Sigma/Merck-Millipore | #05-432 |
| Synaptophysin | Sigma/Merck-Millipore | #S5768 |
| CaMKIIα | Sigma/Merck-Millipore | C6974 |
| Actin | Sigma/Merck-Millipore | MAB1501R |
| GluA2 | Thermo Scientific | #32-0300 |
| PSD95 | Thermo Scientific | MA1-046 |
| GluN2A | Neuromab | #75-288 |
| GluA1 | Abcam | ab31232 |
| RasGRF1 | Santa Cruz | #377234 |
| Homer 1 | Synaptic Systems | #160-004 |
| Rab5 | BD Biosciences | #610281 |
| GFP | Roche | #11814460001 |
| mCherry | GeneTEX | GTX59788 |
| Puromycin | DSHB | PMY-2A4 |
| HRP-conjugated anti-rabbit | Jackson ImmunoResearch | #711-035-152 |
| HRP-conjugated anti-mouse | Jackson ImmunoResearch | #715-035-151 |
| IRDye 680RD goat anti-mouse | LI-COR Biosciences | P/N 926-68070 |
| IRDye 800CW goat anti-rabbit | LI-COR Biosciences | P/N 926-32211 |
| Alexa 647 Donkey anti-Mouse IgG | Thermo Fisher | A-31571 |
| **Oligonucleotides and other sequence-based reagents** | | |
| PCR primers | Eugenio Santos Lab (Esteban et al, 2001) | Sequences in Methods/Animals |
| **Chemicals, Enzymes and other reagents** | | |
| DAPI | Sigma/Merck-Millipore | #10236276001 |
| (RS)-3,5-DHPG | Tocris | #0342 |
| KBSRC4 | Tocris | #4660 |
| Forskolin | Sigma/Merck-Millipore | F6886 |
| Picrotoxin | Sigma/Merck-Millipore | P1675 |
| Rolipram | Sigma/Merck-Millipore | R6520 |
| NMDA | Sigma/Merck-Millipore | M3262 |
| DL-APV | Sigma/Merck-Millipore | A5282 |
| 2-chloroadenosine | Sigma/Merck-Millipore | C5134 |
| Puromycin | Sigma/Merck-Millipore | P8833 |
| PP2 | Sigma/Merck-Millipore | P0042 |
| GF109203 | Sigma/Merck-Millipore | B6292 |
| KN93 | Sigma/Merck-Millipore | K1385 |

| Reagent/Resource | Reference or Source | Identifier or Catalog Number |
|---|---|---|
| **Software** | | |
| AnyMaze 5.14 | Stoelting | |
| pClamp 9.2, 10.3, 11.2 | Molecular Devices | |
| Fiji v1.53t | ImageJ | |
| Image Studio Lite v5.2 | LI-COR Biosciences | |
| NIS Elements 5.30 | Nikon Instruments | |
| GraphPad Prism v8.3 | Dotmatics | |
| **Other** | | |
| Fear conditioning chamber | Coulbourn Instruments | |
| Quintessential stereotaxic injector | Stoelting | 53311 |
| Multiclamp 700 A/B amplifiers | Molecular Devices | |
| Ras-activation assay kit | Cytoskeleton | BK008 |
| Odyssey infrared imager | LI-COR Biosciences | |
| ImageQuant™ LAS 4000 mini | GE Healthcare Life Sciences | |
| Nikon A1R+ confocal microscope | Nikon Instruments | |

## Animals and ethics statement

All biosafety procedures and animal care protocols used were approved by the bioethics committee of CSIC and the local regulatory authorities (PROEX 227/19, 303/19, 160.1/21), and performed according to Spanish (RD 53/2013, 32/2007) and EU guidelines set out in the European Community Council Directives (86/609/EEC). All personnel involved in animal care and experimentation was appropriately trained according to FELASA standards and followed ARRIVE guidelines. Animal's health and welfare were monitored by a designated veterinarian, who determined the humane sacrifice and exclusion from the study of the animals whose welfare was compromised. Animals were housed in cages that meet all regulatory requirements and the animal rooms have a management system that controls temperature, light (12:12 cycle), and humidity. Food and water were provided ad libitum.

Wistar rats of both genders were used to prepare organotypic hippocampal slice cultures.

H-Ras KO mice were generated by substituting *H-Ras* exons I, II and III with a Neomycin cassette in a C57BL/6N genetic background (Hras^tm1Esn, MGI:2385945) (Esteban et al, 2001). This transgenic mouse line was maintained by crossing heterozygous mice, while homozygous WT and KO offsprings of both genders were used for experiments. Genotyping was carried out by polymerase chain reaction (PCR) using the following primers: ATAGTTGTAGGTTGCACCCACATGCCG

(LM88), ACCTGCCAATGAGAAGCACACTTAGCC (LM89), and CTACCGGT GGATGTGGAATGTGTGCGA (LM82).

K-Ras^lox/lox mice were generated by flanking K-Ras exon 1 with two loxP sequences (Drosten et al, 2010). The transgenic mouse line was maintained in homozygosis in a C57BL6-129Sv mixed background.

## Behavior

Behavioral tests were conducted with 2–3 months old H-Ras KO and WT littermates. After handling the animals for 4 consecutive days, tests were performed on different days and following the order described below. Separate cohorts were used for fear conditioning tests and the rest of the tests. Unless otherwise indicated, test cages and objects were cleaned thoroughly with 70% ethanol between subjects and phases to eliminate any olfactory cues. For all behavioral assays, animals were recorded from a camera located above the testing box and analysis was performed manually using the AnyMaze software 5.14 (Stoelting). Analysis was generally performed by an experimenter blind to the genotype and animal order testing randomly assigned.

## Open field test

Open field test was used to evaluate general locomotor activity and anxiety. Mice were placed individually in the center of an acrylic plexiglas box (40 × 40 × 40 cm) with three white opaque walls (two of them with spatial cues) and one black. The animals were allowed to freely explore the arena for 15 min while being recorded.

## Object recognition

This test was performed 24 h after the open field test, which was used as habituation phase. In the first phase (familiarization), mice were placed in the open field arena containing two identical objects in terms of color, texture and shape on two adjacent corners of the arena, randomly assigned. Mice were allowed to explore the objects for 10 min while being recorded. In the second phase (reexposure; 1 h later), mice were placed again in the arena containing the same objects in the same position, and were again recorded for 10 min. Finally, in the novel object phase of the test (1 h later), one of the objects was replaced for another one of similar dimensions and texture but different in color and shape. Then, mice were allowed to explore both objects for 10 min. The objects used were cylindrical jars filled with aluminum foil and rounded water feeding bottles filled with bedding material. Exploration of the objects was defined as close contact sniffing (with the head pointed towards the object and within 2 cm of the object). Discrimination index was calculated as the time spent exploring the novel object minus the time exploring the familiar object, divided by the time exploring both objects. Mice that did not explore the two objects for at least 10 s were excluded from analyses.

## Sociability and social memory tests

We used a three-chamber acrylic box with dividing walls and a gate to give the testing mouse access to each chamber. The test consisted of three phases. During the habituation phase, empty

wire cages were placed on the two outermost chambers allowing visual, auditory, olfactory, and minimal touch interaction with the inside. Mice were allowed to explore the arena for 5 min. The second phase, the sociability task, was performed 24 h later. An unfamiliar mouse (gender and age-matched with the subject) was placed under the wire cage of one of the chambers and an inanimate object was placed on the opposite cage in an identical configuration. The unfamiliar mice had been habituated to the wire cage for intervals of 10 min the 2–3 days prior to the test. The subject mouse was placed in the center chamber and allowed to freely explore all three chambers for 10 min. The test mouse was removed and put in a separate home cage for 5 min while cleaning the arena and the wire cages. For the third phase, the social memory task, the inanimate object was replaced by another unfamiliar mouse and the test was performed as in the second phase, putting back into the arena the tested mouse for another 10 min session. Exploration of the targets was defined as close (within 2 cm) contact sniffing of the wire cage (with the head pointed towards the cage). Sitting or standing time was not quantified as exploration. Discrimination index was calculated as the time exploring the subject minus the object divided by the time exploring both of them (first phase), and the time exploring the new subject minus the familiar one divided by the time exploring both (second phase). Mice that did not explore the two subjects for at least 10 s were excluded from analyses.

## Y maze test

The Y maze test was performed to quantify spontaneous alternation and assess spatial working memory. The apparatus consists of a Y-shaped maze with three identical enclosed arms. To perform the test, we placed each mouse in the center of the maze and recorded a single trial of 5 min. The sequence of arm entrance was analyzed manually, considering an alternation as a consecutive entrance to all three arms. The spontaneous alternation percentage was calculated using the following equation:

$$\% \, Spontaneous \, alternation = \frac{number \, of \, alterantions}{total \, number \, of \, arm \, entrances - 2} \, x \, 100$$

## Contextual and cued fear conditioning tests

The tests consisted of a training session followed by a test session 24 h later. During the training session (common to both tests), mice were placed in a fear conditioning chamber ($25 \times 31 \times 25$ cm, Coulbourn, Allentown, PA) with aluminum and Plexiglas walls and a floor consisting of stainless-steel bars (26 parallel steel rods 5 mm diameter, 6 mm spacing) that can be electrified to deliver a mild shock. The chamber was located in a sound attenuating cubicle (Med Associates, Burlington, VT) with a ventilating fan, which produced an ambient noise level of 58 dB. A speaker was mounted on the wall, and illumination was provided by a single overhead light (miniature incandescent white lamp 28 V). After 2 min of habituation, three tone-footshock pairings were delivered. The 80 dB, 4 kHz tone lasted for 30 s and the footshocks were 0.75 mA and lasted 2 s, co-terminating with the tone. Inter-tone interval lasted for 30 s and mice remained in the chamber for 30 s after the last shock.

Twenty-four hours after the training, a cohort of mice was tested for contextual fear memory. These mice were re-introduced in the same fear conditioning chamber for 5 min and freezing behavior was analyzed (see below). A different cohort of mice was tested for cued fear memory also 24 h after training. In this case, the chamber was physically distinct from the training chamber, as it had a triangular shape with a white smooth plexiglas floor. The odor changed from ethanol 70% to acetic acid 5%. Mice were placed in the new context and after 2 min of initial exploration they were re-exposed to the tone paired with the shock in the training session. The tone lasted for 5 s with 25 s inter-tone interval. Fourteen repetitions of the tone were presented, and mice remained in the chamber for 25 s after the last tone.

Memory was assessed by the duration of freezing behavior. The freezing response is defined as the absence of movement in the mouse except for the breathing movements. Data was presented as percentage of time freezing when re-exposed to the context or the tone previously paired with the aversive stimulus.

## Stereotaxic in vivo injections

AAV5-CaMKII-Cre-mCherry virus (serotype 5, $5.8 \times 10^{12}$ viral particles/mL, Gene Therapy Vector Core at the University of North Carolina) was delivered to the hippocampus of K-Ras$^{lox/lox}$ mice through two bilateral injections (600 nL each, 120 nL/min) in the dorsal (in mm from bregma, AP: −1.8, ML: ±1.3, DV: −1.7) and ventral (AP: −2.8, ML: ±2.4, DV: −2) regions. The 2–3-month-old animals were placed in the stereotaxic frame (Harvard Apparatus) over a heating pad and immobilized using blunt ear holders. Anesthesia was initiated and maintained by inhalation of isoflurane 2% in oxygen. Viral solution was infused using a Hamilton Neuros syringe (65460-02) inserted through small holes drilled into the skull and coupled to a micropump (Quintessential stereotaxic injector, Stoelting-53311). After the infusion, the syringe was kept in place for 5 min and slowly withdrawn. After the closure of the wound, the animals received analgesia by subcutaneous injection of Meloxicam (5 mg/kg dose). One month after the surgery and before performing electrophysiology experiments, the accuracy of the infection was confirmed by immunofluorescence using the expression of mCherry reporter, observing mCherry-Cre-expressing neurons along the entire hippocampus.

## Acute slice preparation

Acute hippocampal slices were obtained from 3–4-month-old mice of both genders. Animals were anesthetized with isoflurane and quickly decapitated once they were irresponsive to tail and foot pinches. The brains were rapidly removed and submerged in $Ca^{2+}$ free ice-cold dissection solution (10 mM D-glucose, 4 mM KCl, 26 mM $NaHCO_3$, 233.7 mM sucrose, 5 mM $MgCl_2$, 0.001% [w/v] phenol red as a pH indicator) gassed with carbogen (5% $CO_2$/95% $O_2$). Three hundred and fifty μm-thick coronal slices were obtained cutting the brain in the same dissection solution using a vibratome (Leica, VT1200S). Slices were left in carbogen-gassed artificial cerebrospinal fluid (aCSF) (119 mM NaCl, 2.5 mM KCl, 1 mM $NaH_2PO_4$, 26 mM $NaHCO_3$, 11 mM glucose, 1.2 mM $MgCl_2$, 2.5 mM $CaCl_2$, osmolarity adjusted to $290 \pm 5$ mOsm) for 1 h at

32 °C to recover and then were maintained at 25 °C until used for electrophysiological field recording experiments.

## Organotypic hippocampal slice cultures

Organotypic cultures were prepared from 5- to 7-day-old Wistar rat pups of both sexes (or H-Ras WT and KO mice when indicated) according to previously described procedures (Gähwiler, 1997). Briefly, whole brains were extracted and immersed in ice-cold dissection medium (10 mM D-glucose, 4 mM KCl, 20 mM NaHCO$_3$, 233.7 mM sucrose, 5 mM MgCl$_2$, 1 mM CaCl$_2$ with 0.001% [w/v] phenol red as pH indicator) gassed with carbogen. Hippocampi were dissected under sterile conditions and 400-μm-thick slices were prepared using a McIlwain tissue chopper. Individual slices were separated and placed in culture on porous membranes (PICM0RG50, Merck Millipore) over culture medium (0.8% [w/v] MEM powder, 20% [v/v] horse serum, 1 mM L-glutamine, 1 mM CaCl$_2$, 2 mM MgSO$_4$, 1 mg/L insulin, 0.0012% [v/v] ascorbic acid, 30 mM HEPES, 13 mM D-glucose, 5.2 mM NaHCO$_3$). Slices were maintained in vitro at 35.5 °C and 5% CO$_2$ for 6–11 days until use, replacing the culture medium every 2–3 days. Electrophysiological or biochemical experiments were performed in carbogen-gassed aCSF) (119 mM NaCl, 2.5 mM KCl, 1 mM NaH$_2$PO$_4$, 26 mM NaHCO$_3$, 11 mM glucose, 4 mM MgCl$_2$, 4 mM CaCl$_2$, osmolarity adjusted to 290 ± 5 mOsm).

## DNA constructs and expression of recombinant proteins

GFP-Ras construct was made by in-frame ligation of H-Ras sequence into pEGFP-C1 vector. Ras-DN (S17N) and Ras C181,184S mutants were generated from WT H-Ras sequence by mutagenic PCR. FRas2-M construct was purchased from Addgene (#45149). Constructs were expressed for 12–16 h in organotypic hippocampal slices, using Sindbis virus or biolistics transfection. For electrophysiology and biochemical experiments, infection was performed after 7–10 DIV and it was restricted to CA1 area. For FRET imaging experiments, slices were biolistically transfected after 14–18 DIV.

## Electrophysiology

Excitatory postsynaptic currents (EPSCs) and field excitatory postsynaptic potentials (fEPSPs) were recorded from CA1 pyramidal neurons with glass recording electrodes while stimulating Schaffer collateral fibers with Platinum-iridium bipolar cluster microelectrodes (FHC, CE2C55). During the recordings, the slices were placed in an immersion chamber constantly perfused with aCSF gassed with 5% CO$_2$ and 95% O$_2$ and its temperature was maintained at 29 °C for whole-cell recordings and mGluR-LTD field recordings, and at 25 °C for NMDAR-LTP field recording experiments. For patch-clamp recordings, glass electrodes (4–6 MΩ) were filled with intracellular solution composed of 115 mM CsMeSO$_3$, 20 mM CsCl, 10 mM HEPES, 2.5 mM MgCl$_2$, 4 mM Na$_2$-ATP, 0.4 mM Na-GTP, 10 mM sodium phosphocreatine, 0.6 mM EGTA, 10 mM Lidocaine N-ethyl bromide, pH adjusted to 7.25 and osmolarity to 290 ± 5 mOsm. Unless otherwise indicated, all whole-cell recordings were performed at −60 mV, measuring AMPAR-mediated responses. NMDAR-mediated responses were measured at +40 mV, 65 ms after

stimulation, when AMPAR-mediated responses had fully decayed. For field recordings, glass pipettes (0.5–1 MΩ) were filled with aCSF, used as extracellular solution. Stimulation intensity was adjusted to 30% for NMDAR-LTP or 50% for mGluR-LTD of the maximum slope.

mGluR-LTD was induced using low frequency (1 Hz, 900 pulses) presynaptic stimulation with paired pulses (50 ms inter-stimulus interval) (PP-LFS). NMDAR-dependent LTP was induced with a theta-burst stimulation protocol (TBS) composed of 10 trains of bursts (4 pulses at 100 Hz with a 200 ms interval), repeated for 4 cycles with 20 s inter-cycle interval. For paired-pulse ratio (PPR) measurements, the first 5 pulse pairs (50 ms inter-stimulus intervals, 1 Hz) of mGluR-LTD induction were used. PPR was calculated by dividing the peak amplitude of the second response by the peak amplitude of the first.

For all experiments, aCSF was supplemented with 100 μM picrotoxin to block GABA$_A$ receptors. In addition, for mGluR-LTD experiments 100 μM DL-APV was added to the aCSF to block NMDARs. Four μM 2-chloroadenosine was added in patch-clamp recordings, and 5 μM PP2 or 10 μM KBSRC4 were added when indicated. Electrophysiological recordings and data acquisition were achieved using Multiclamp 700 A/B amplifiers and pClamp software 9.2, 10.3, 11.2 (Molecular Devices). Data analysis was performed with custom made Excel (Microsoft) macros.

## Pharmacological treatments

Hippocampal organotypic slices were transferred to a submersion-type holding chamber containing aCSF gassed with carbogen at 29 °C. After 10 min of acclimation, chemical stimulation of synaptic plasticity was induced by 100 μM DHPG, cLTP (0.1 μM rolipram, 50 μM forskolin and 100 μM picrotoxin) or 20 μM NMDA stimulation for 10, 10 and 5 min respectively. In biochemical experiments following viral infection, the CA1 region was dissected and homogenized after DHPG stimulation. When using PP2 (5 μM), KN93 (20 μM) or GF109203 (200 nM) inhibitors, acclimation lasted 30 min and inhibitors were present also during DHPG stimulation.

## Puromycin labeling

Basal protein translation was measured with the SUnSET assay, as previously described with modifications (López-Merino et al, 2023; Schmidt et al, 2009). Briefly, cultured hippocampal slices, infected the day before with a mix of Sindbis viruses expressing Ras-DN and TdTomato, were treated with puromycin (1 μg/mL) ±DHPG in aCSF for 10 min. After treatment, slices were kept in aCSF for 20 min and then fixed with 4% paraformaldehyde (PFA) and 4% sucrose in PBS pH 7.4 for 1 h at room temperature. Slices were then blocked (3% BSA, 3% horse serum, and 0.1% Triton X-100) and immunostained with anti-puromycin antibody. Nuclei were labeled with DAPI after immunostaining. Samples were mounted onto microscope slides and fluorescence images were acquired as 8-μm-depth Z-stacks starting from the slice surface using confocal microscope Nikon A1R+ with a 60× 1.4 oil Apochromat objective. All images were acquired using the same microscope settings and conditions. From 6- to 8-μm-depth Z-stacks were reconstructed (maximum intensity projection) and analyzed using the Fiji v1.53t software (ImageJ). Puromycin labeling was quantified in CA1 neurons infected with either Ras-DN-GFP or TdTomato, and in

uninfected neurons (using DAPI as reference). Neurons coinfected were sparse and were not used for analysis.

## Ras-GTP pull-down

The activity of Ras was assessed using the Cytoskeleton Ras-activation assay biochem kit (BK008, Cytoskeleton), according to the manufacturer's instructions. Briefly, after the corresponding pharmacological treatments, slices were lysed using the provided lysis buffer. Equal amounts of proteins (~1 µg/mL) were added to the effector-coupled beads provided, and lysates with beads were incubated on a rotator wheel at 4 °C for 1 h. Beads were then centrifuged at $1000 \times g$ for 1 min, washed with the provided wash buffer, and resuspended in 2X sample buffer. Western blotting was performed using the provided antibodies. Data is presented as a ratio of GTP-bound (pull-down) to total Ras (input) in each experimental sample. For all pull-down assays, ratios were normalized to the control sample.

## FRET live imaging experiments: Ras activity reporter

Live imaging experiments were performed on CA1 neurons co-transfected biolistically with H-Ras-GFP and FRas2-M sensor (Oliveira and Yasuda, 2013), focusing on dendritic spines from secondary/tertiary dendritic segments. Organotypic hippocampal slices were placed in a chamber continuously perfused with aCSF gassed with carbogen (5% $CO_2$/95% $O_2$) at 29 °C. After baseline, mGluR-LTD was induced chemically with 100 µM DHPG for 10 min, followed by a wash out with aCSF. Confocal fluorescence images were acquired as Z-stacks with a confocal microscope Nikon A1R+ using a 60x NA 1.2 water Apochromat objective, a 3x zoom factor and 488-nm, 561-nm, and 640-nm lasers in combination with NIS Elements 5.30 software. All images were acquired using the same microscope settings and conditions. For image analysis and quantification, Z-stacks were reconstructed (maximum intensity projection) using the software Fiji v1.53t (ImageJ). FRET signal was corrected by using the following equation:

$$FRET' = S_{FRET} - F_A * S_A - F_D * S_D$$

where $S_{FRET}$, $S_A$, and $S_D$ are the raw signals obtained from FRET, acceptor and donor channels, respectively, and

$$F_A = \frac{S_{FRET}}{S_A}$$

$$F_D = \frac{S_{FRET}}{S_D}$$

which were obtained from slices transfected only with FRas2-M sensor (acceptor) or H-Ras-GFP (donor). Donor fluorescence signal that decreased over 25% of its baseline was considered to be bleached and excluded from analyses.

## Subcellular fractionation experiments

Subcellular fractionation of hippocampal tissue was performed as previously reported (Briz et al, 2017) with minor modifications.

Briefly, both hippocampi were pooled together and homogenized in sucrose buffer (0.32 M sucrose, 1 mM HEPES, 1 mM $MgCl_2$, 1 mM EDTA, 1 mM $NaHCO_3$ at pH 7.4). Alternatively, acute hippocampal slices were stimulated with DHPG for 10 min, and 9–12 slices were pooled together and homogenized as before. The homogenized tissue was centrifuged at $1000 \times g$ for 10 min. The resulting supernatant (S1) was centrifuged at $13,000 \times g$ for 15 min to obtain a crude membrane fraction (P2). The pellet was resuspended in buffer containing 75 mM KCl and 1% Triton X-100 and centrifuged at $100,000 \times g$ for 1.5 h. The supernatant (S3) was referred to as non-PSD fraction. The final pellet (P3) was homogenized in a glass-glass potter in 20 mM HEPES, 0.1% NP-40 and referred to as PSD fraction. S2 fraction was also centrifuged at $100,000 \times g$ for 1.5 h. The resulting pellet (P2*) was homogenized in 20 mM HEPES, 0.1% NP-40 and referred as microsomal fraction. All purifications were performed in presence of protease inhibitors (Complete tablets, #04693132001, Roche) and phosphatase inhibitors (pSTOP, #04906837001, Roche). Samples were stored at −80 °C until used for western blotting.

## Western blotting

Protein extracts were processed by SDS–polyacrylamide gel electrophoresis and transferred to polyvinylidene fluoride (PVDF) membranes (Immobilon-P, Merck Millipore). Afterward, membranes were blocked (5% w/v non-fat dry milk in Tris-buffered saline + 0.1% Tween-20) (1 h at room temperature) and incubated with the corresponding primary antibodies. Immunodetection was done either by chemiluminescence with 1–5 min ECL incubation (Enhanced ChemiLuminiscence, Immobilon Western, Millipore) and the ImageQuant™ LAS 4000 mini biomolecular imager (GE Healthcare Life Sciences), or by IRDye-conjugated secondary antibodies and Odyssey infrared imager (LI-COR Biosciences). Image Studio Lite v5.2 software (LI-COR Biosciences) was used to analyze and quantify signal intensities obtained from digital images.

## Statistical analysis

Sample size estimations for each experiment were based on similar experiments from our previous studies (Sánchez-Castillo et al, 2022; López-Merino et al, 2023). Results were calculated as means ± standard error of the mean (SEM). The number of independent experiments ($n$) for electrophysiology experiments refers to the number of neurons for patch-clamp recordings (obtained from at least 4 animals) or slices for field recordings (obtained from at least 9 animals). For biochemical experiments, unless otherwise indicated, $n$ refers to the number of independent cultured batches (each batch was a combination of slices from 2 pups). For behavioral experiments, $n$ refers to the number of animals used.

Unless otherwise indicated, statistical differences between groups were determined via different non-parametric tests, the cut-off value for statistical significance was set at $p$-value $< 0.05$. Mann-Whitney U test was used for unpaired data, Wilcoxon test for paired data and one-way ANOVA/Kruskal–Wallis + Dunn's post-test or two-way ANOVA/Mixed effects model + Sidak's post-test for comparing multiple groups. All statistical analysis and graphic representations were performed using GraphPad Prism software v8.3. Criteria for outlier exclusion was based on Rout test (Q = 1%) using GraphPad Prism.

## Data availability

Our study includes no data deposited in public repositories.

The source data of this paper are collected in the following database record: biostudies:S-SCDT-10_1038-S44318-025-00390-8.

## Peer review information

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

## Acknowledgements

We thank Irene Varela for her preliminary work with Ras mutants. We also thank the members of the Esteban laboratory for critical reading of the manuscript and the personnel at the fluorescence microscopy facility (SMOA) and the animal house of the CBM for expert technical assistance. This work was supported by the Spanish Ministry of Science, Innovation and Universities: grants PID2020-117651RB and PDC2021-120815-I00 to JAE, grant PID2020-119358GB-I00 to DFS, and grants PID2022-136932OB-I00 and RYC2021-031395-I to VB. CG was supported by grants from Fondo de Investigación Sanitaria (Project PI11-02529) and Fundación Ramón Areces (FRA 01-09-001). ELM was supported by a predoctoral FPU18/02838 contract from the Spanish Ministry of Science.

## Author contributions

**Esperanza López-Merino**: Conceptualization; Data curation; Formal analysis; Investigation; Methodology; Writing—original draft. **Alba Fernández-Rodrigo**: Methodology. **Jessie G Jiang**: Formal analysis; Investigation. **Silvia Gutiérrez-Eisman**: Methodology. **David Fernández de Sevilla**: Methodology. **Alberto Fernández-Medarde**: Resources. **Eugenio Santos**: Resources. **Carmen Guerra**: Resources. **Mariano Barbacid**: Resources. **José A Esteban**: Conceptualization; Supervision; Funding acquisition; Project administration; Writing—review and editing. **Víctor Briz**: Conceptualization; Formal analysis; Supervision; Funding acquisition; Investigation; Methodology; Project administration; Writing—review and editing.

Source data underlying figure panels in this paper may have individual authorship assigned. Where available, figure panel/source data authorship is listed in the following database record: biostudies:S-SCDT-10_1038-S44318-025-00390-8.

## Disclosure and competing interests statement

The authors declare no competing interests.

# Expanded View Figures

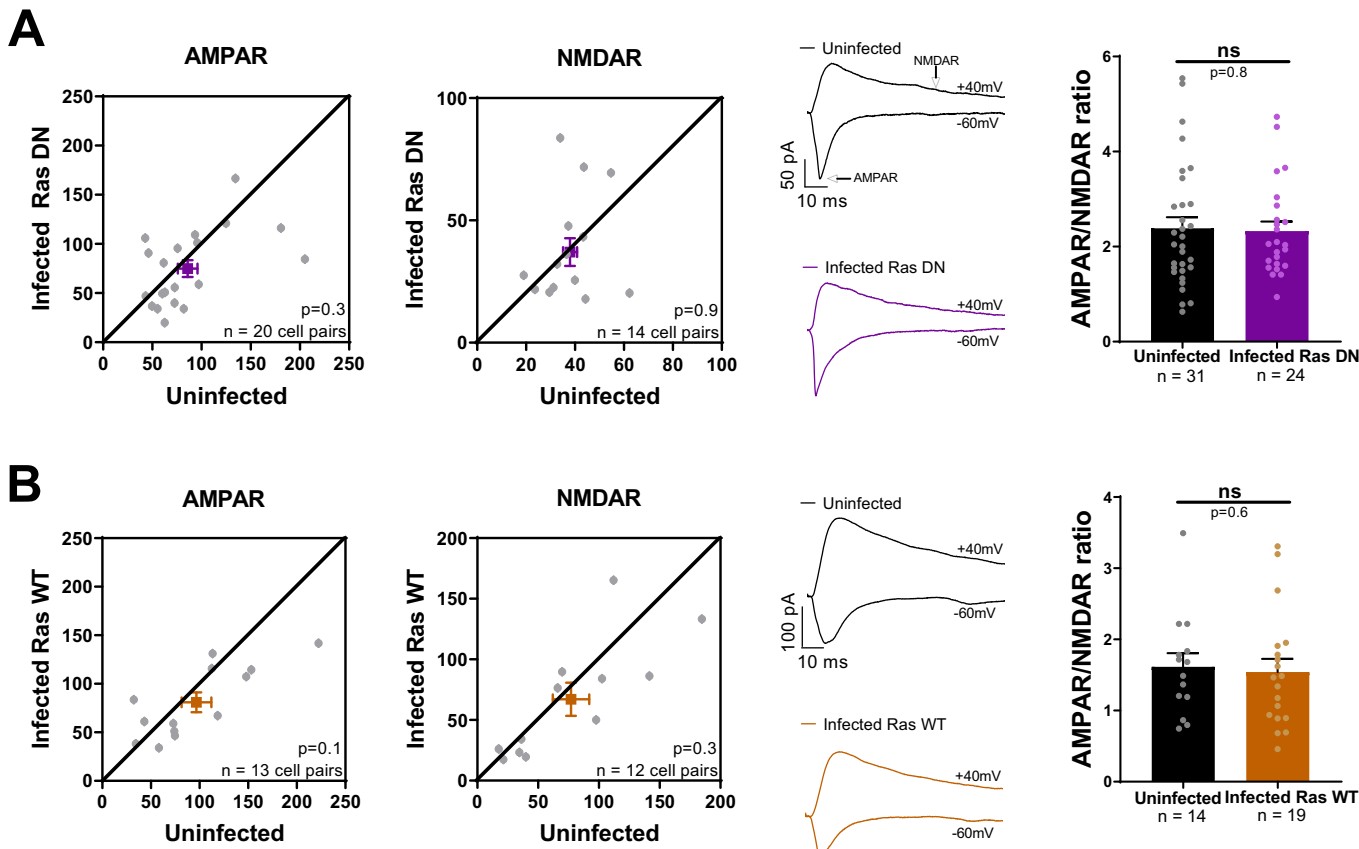

**Figure EV1.  Effect of Ras dominant negative on basal synaptic transmission.**

Scatter plot of simultaneous recordings of AMPAR- or NMDAR-mediated responses from uninfected and Ras-DN (**A**) or Ras WT (**B**) infected neurons (left). Representative traces (middle). Quantification of AMPA/NMDA ratio (right) represented as mean ± SEM; individual values are also plotted. Mann-Whitney test was used to evaluate significant differences. ns, non-significant.

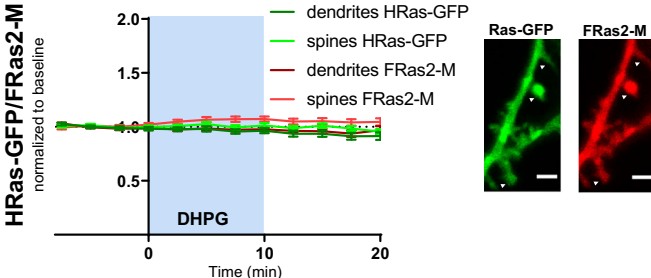

**Figure EV2.    H-Ras-GFP and FRas2-M fluorescence during DHPG treatment.**

Left. Time course of H-Ras-GFP and FRas2-M signal in spines ($n = 50$) and dendrites ($n = 15$) of CA1 pyramidal cells ($n = 10$) upon DHPG stimulation. Results are represented as mean ± SEM. Kruskal–Wallis test was used to evaluate differences across time. [$H(11,571) = 15.1$, $p = 0.2$ for FRas2-M and $H(11,571) = 18.8$, $p = 0.06$ for Ras-GFP]. Right. Representative images of dendritic branches of CA1 neurons coexpressing H-Ras-GFP (green channel) and FRas2-M (red channel) constructs. Scale bars 2 μm. White arrows point dendritic spines.

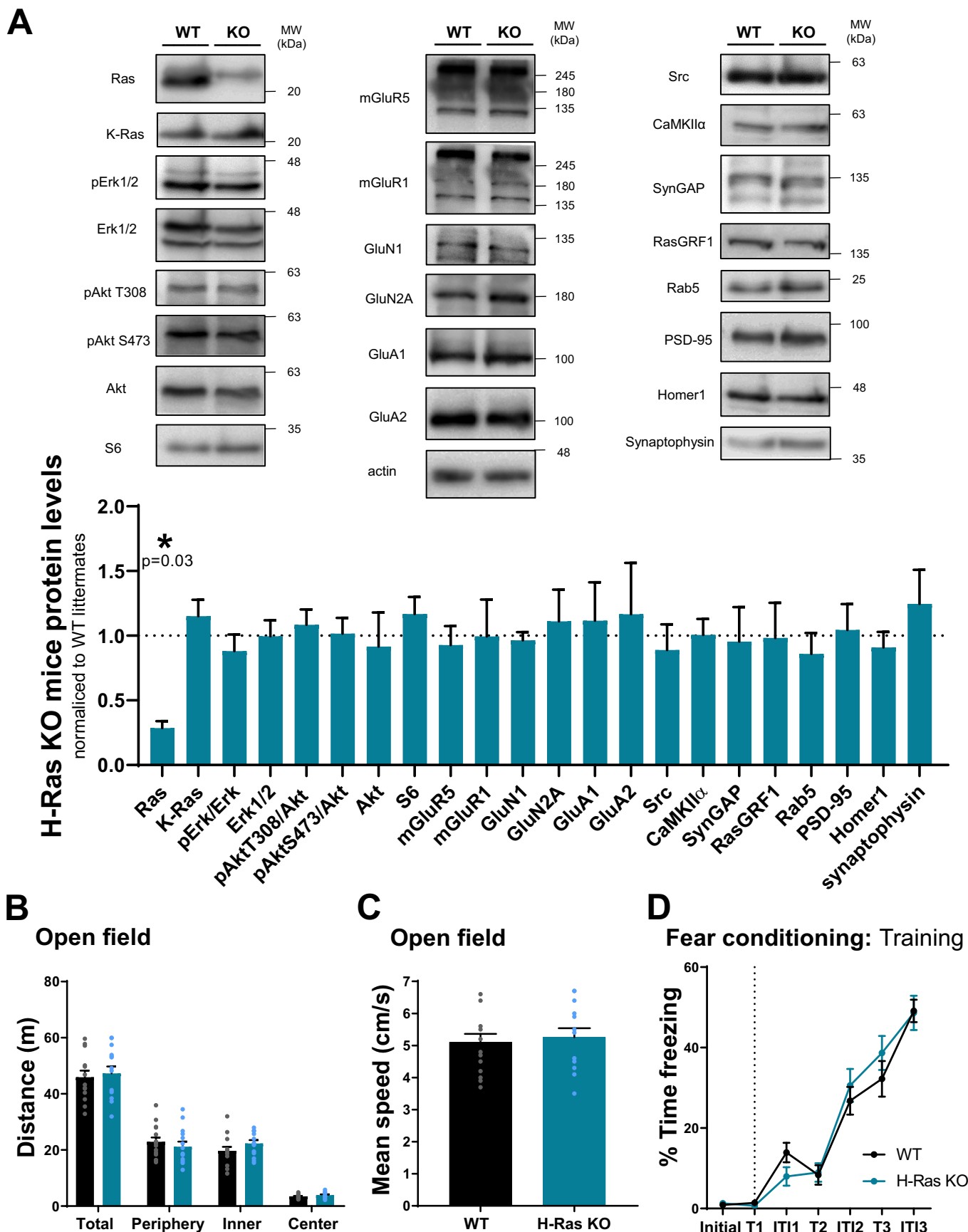

◀ **Figure EV3.** **Hippocampal levels of different synaptic and signaling proteins and animal behavior in H-Ras KO mice.**

(A) Representative Western blots (upper) and quantification (lower) of Ras proteins and different Ras effectors, signaling molecules and glutamate receptors. Some blots have been reused from Fig. 5 (inputs of K-Ras, Rab5, PSD-95, synaptophysin), as they come from the same experiment/animal. Unless otherwise indicated, each sample is normalized to actin levels and results are referred to WT animals and represented as mean ± SEM, $n = 6$ animals per condition. Wilcoxon signed-rank test (*) was used to evaluate changes; non-significant differences were found except for total Ras levels. (B) Total distance traveled in the open field test and in the different subsections of the arena. Mean ± SEM; individual values are also represented. Mixed effects analysis was used to evaluate differences between genotypes [$F_{(1,27)} = 0.2$, $p = 0.7$]. (C) Mean speed in the open field test. Mean ± SEM; individual values are also represented. Mann-Whitney test was used to evaluate differences between genotypes ($p = 0.9$). (B, C) $n = 15$ mice for WT and 14 mice for H-Ras KO. (D) Percentage of time spent freezing during fear conditioning training session. T, tone; ITI, inter-tone interval. Mean ± SEM is represented. Mixed effects analysis was used to evaluate differences between genotypes [$F_{(1,45)} = 0.1$, $p = 0.8$]. $n = 25$ mice for WT and 22 mice for H-Ras KO.

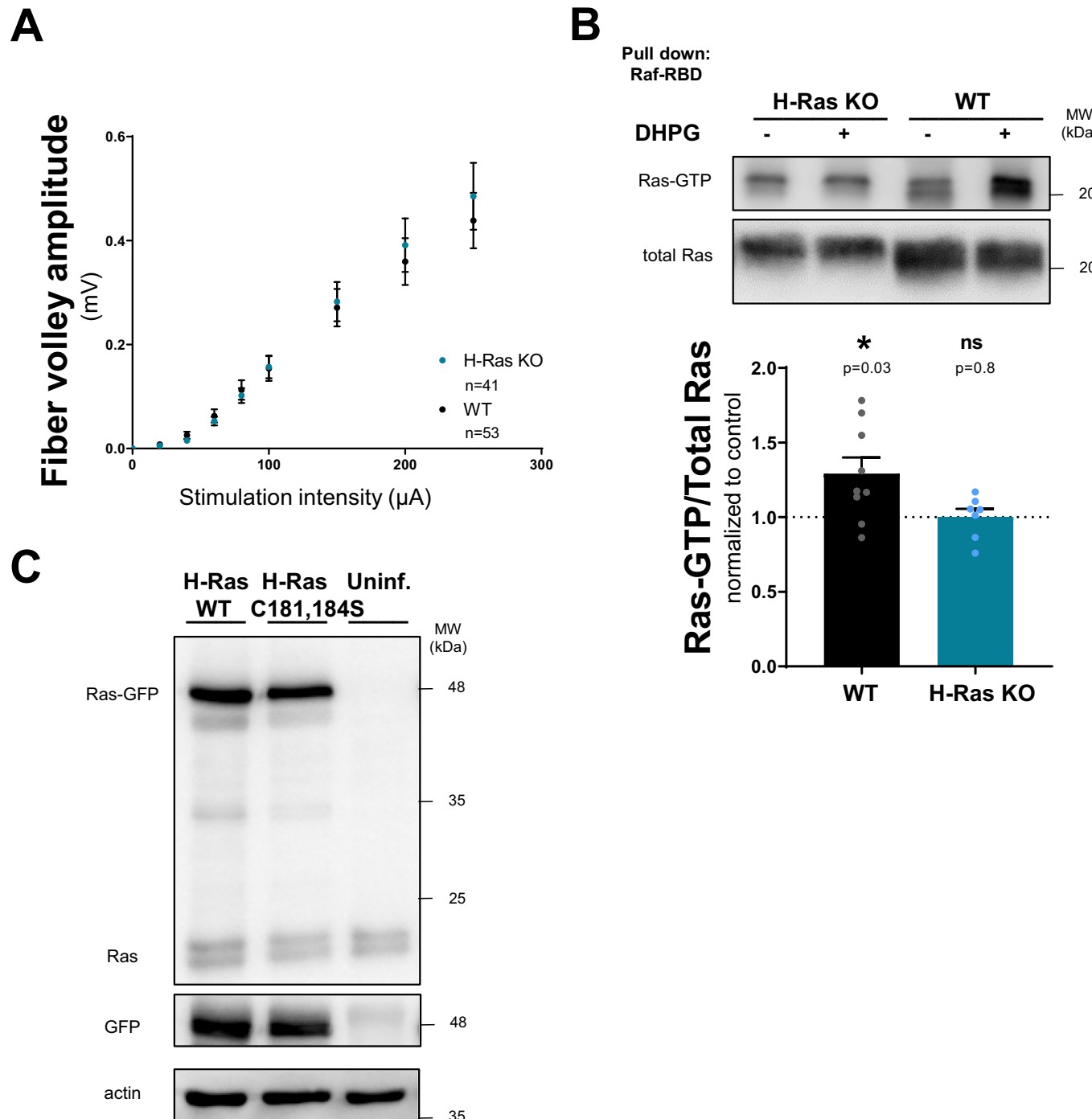

**Figure EV4. Analysis of basal transmission and Ras activity in H-Ras KO mice.**

(A) Fiber volley responses over increasing stimulation intensities during input/output curve measurements shown in Fig. 4A. Mean ± SEM; Mixed effects analysis [$F_{(1,92)} = 0.1$, $p = 0.8$]. (B) Quantification (lower panel) and representative blot (upper panel) of active Ras pulled-down vs total Ras from acute hippocampal slices of WT and H-Ras KO mice 10 min after DHPG treatment. Results are normalized to control non-stimulated slices (in WT or KO mice) and expressed as mean ± SEM, $n = 7$–9 independent slices/mice. Wilcoxon signed-rank test (*) was used to assess statistically the effect of DHPG. ns, non-significant. (C) Representative blot of Ras and GFP in organotypic hippocampal slices infected with H-Ras WT and H-Ras C181,184S, or uninfected (uninf.). Note that both endogenous Ras (~21 KDa) and recombinant Ras-GFP (~48 KDa) were detected at their appropriate sizes. Actin was used a protein loading control.

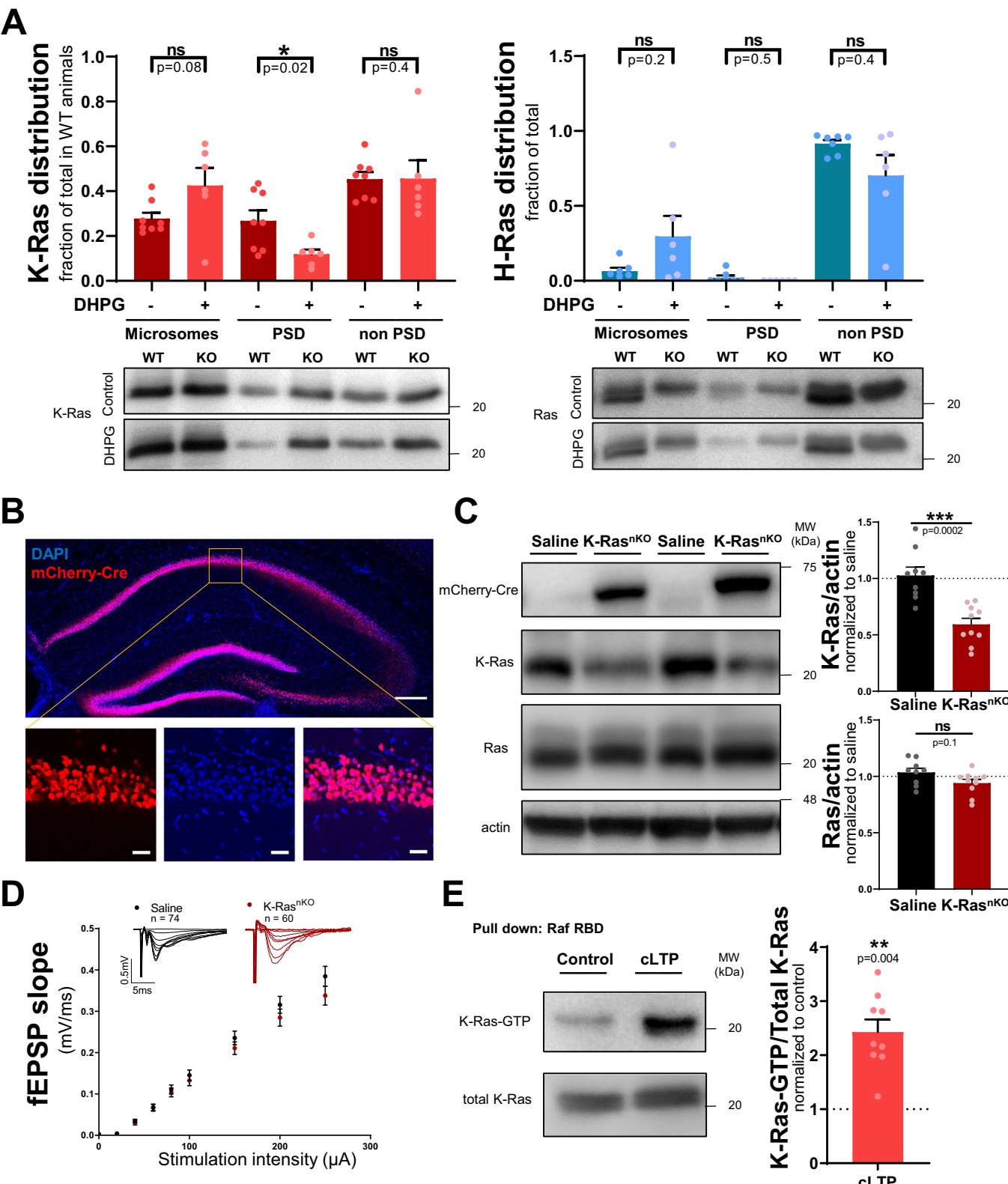

**Figure EV5.   Analysis of Ras distribution, expression, and activity in H-Ras and neuronal K-Ras KO mice.**

(A) Representative blots (lower pannel) of total Ras and K-Ras in different subcellular compartments following DHPG stimulation in hippocampal slices from WT and H-Ras KO mice. H- and K-Ras distribution in microsomes, PSD and non-PSD fractions was quantified (upper panel) as percentage of total protein (from inputs) for each isoform. H-Ras signal was obtained by subtracting total Ras signal of H-Ras KO mice from that of WT mice. Results expressed as mean ± SEM, $n = 6$–8 animals per condition (controls from Fig. 5A were included in the quantification); individual values are also represented. Mann-Whitney test was used to evaluate DHPG effect in each fraction compared to control, non-stimulated slices. (B) Representative image of mCherry fluorescence (red) and DAPI staining (blue) of a hippocampal slice from AAV-CaMKII-mCherry-Cre-infected K-Ras$^{flox/flox}$ mice (lower panels: zoom-in images of CA1 region). Scale bars 200 (upper) and 20 (lower) μm. (C) Representative Western blots (left) and quantification (right) of K-Ras and total Ras hippocampal levels. Results are normalized to saline average levels and expressed as mean ± SEM; individual values are also represented. Mann-Whitney test (*) was used to evaluate statistical differences. $n = 9$ (WT) and 10 (K-Ras nKO) mice. (D) Input/output curves of fEPSP slopes vs stimulation intensities. Representative traces are shown in the upper part. Mean ± SEM; Mixed effects analysis was used to assess statistical differences between genotypes [$F(1,132) = 1.0$, $p = 0.3$]. (E) Quantification (left) and representative blot (right) of active K-Ras pulled-down vs total K-Ras from mouse organotypic hippocampal slices 10 min after cLTP treatment. Results are normalized to control non-stimulated slices and expressed as mean ± SEM, $n = 9$. Wilcoxon signed-rank test (*) was used to assess statistically the effect of cLTP induction. ns, non-significant.

