## [Peer Review File · The EMBO Journal]

Different Ras isoforms regulate synaptic plasticity in opposite directions

Esperanza Lopez-Merino, Alba Fernandez-Rodrigo, Jessie Jiang, Silvia Gutiérrez-Eisman, David Fernandez de Sevilla, Alberto Fernandez-Medarde, Eugenio Santos, Carmen Guerra, Mariano Barbacid, Jose Esteban, and Víctor Briz

Corresponding author(s): Víctor Briz (victor.briz@isciii.es) , Jose Esteban (jaesteban@cbm.csic.es)

Review Timeline:

Submission Date:	8th Jul 24
Editorial Decision:	10th Sep 24
Revision Received:	24th Dec 24
Editorial Decision:	21st Jan 25
Revision Received:	27th Jan 25
Accepted:	30th Jan 25

Editor: Ioannis Papaioannou

Transaction Report:

Dear Dr. Briz,

Thank you again for submitting your manuscript EMBOJ-2024-118427 to The EMBO Journal and for your patience during peer review. Your manuscript has now been seen by three experts in the field, and we have received the full set of their comments, which you can find below.

As you will see, all referees are very positive and supportive, they recognize that the presented findings are novel, interesting, and important, they find the experiments well-designed and the data compelling, and they point out that the results are clearly presented in a well-written manuscript. However, they also identify a number of limitations and provide a list of additional suggestions for further improvement of the study and the manuscript, which we would like you to address in a revised version of your manuscript.

Please include in your resubmission a detailed point-by-point response addressing all referees' comments and concerns. I should add that it is EMBO Journal policy to allow only a single round of major revision, and acceptance of your manuscript will therefore depend on the completeness of your responses in this revised version. Please let me know if you have any questions or comments that you would like to discuss further with me.

We generally allow three months as standard revision time (December 9, 2024). As a matter of policy, competing manuscripts published during this period will not negatively impact our assessment of the conceptual advance presented by your study. However, we request that you contact us as soon as possible upon publication of any related work, to discuss how to proceed. Should you foresee a problem in meeting this three-month deadline, please let us know in advance and we may be able to grant an extension.

Thank you for the opportunity to consider your work for publication in The EMBO Journal. I look forward to your revision.

Best regards,

Ioannis

Instructions for preparing your revised manuscript

1. When you are ready to submit the revision, please upload:

- A Word file of the manuscript text (including legends of main Figures, EV Figures and Tables). Please make sure that changes are highlighted (or "tracked") to be clearly visible.

- Individual production-quality figure files (one file per figure). When assembling your figures, please refer to our figure preparation guidelines in order to ensure proper formatting and readability in print as well as on screen:

If the data shown in a figure are obtained from n {less than or equal to} 2, please use scatter plots showing the individual data points.

- i. the name of the statistical test used to generate error bars and P values
- ii. the number (n) of independent experiments (please specify technical or biological replicates) underlying each data point (discussion of statistical methodology can be reported in the Materials and Methods section, but figure legends should contain a basic description of n , P , and the test applied)
- iii. the nature of the bars and error bars (s.d., s.e.m.).

- A point-by-point response to the referees' comments, with a detailed description of the changes made (as a word file). All referees' concerns must be fully addressed and their suggestions taken on board. When preparing your letter of response to the referees' comments, please bear in mind that this will form part of the Review Process File and will therefore be available online

to the community. Please note that you have the possibility to opt out of the transparent process at any stage prior to publication by letting the editorial office know (contact@embojournal.org); if you do opt out, the Review Process File link will point to the following statement: "No Peer Review File is available with this article, as the authors have chosen not to make the review process public in this case.". For more details on our Transparent Editorial Process, please visit our website: <https://www.embopress.org/page/journal/14602075/authorguide#transparentprocess>

- Expanded View (EV) files (replacing Supplementary Information) that are collapsible/expandable online. A maximum of 5 EV Figures can be typeset. EV Figures should be cited as "Figure EV1, Figure EV2" etc. in the text, and their respective legends should be included in the manuscript file after the legends of regular figures. See detailed instructions regarding Expanded View files here:

- For the figures that you do NOT wish to display as Expanded View figures, they should be bundled together with their legends in a single PDF file called "Appendix", which should start with a short Table of Contents (including page numbers). Appendix figures should be referred to in the main text as: "Appendix Figure S1, Appendix Figure S2" etc. Please see detailed instructions here: <https://www.embopress.org/page/journal/14602075/authorguide#expandedview>

- A complete author checklist, which you can download from our author guidelines (<https://www.embopress.org/page/journal/14602075/authorguide>). Please note that the checklist will also be part of the Review Process File.

2. Please note that no statistics should be calculated and shown in Figures if $n=2$. Please also note that each p value should be reported as an exact value.

3. Before submitting your revision, primary datasets (and computer code, where appropriate) produced in this study need to be deposited in appropriate public databases (see <https://www.embopress.org/page/journal/14602075/authorguide#dataavailability>). Their accession numbers, databases, and the specific URLs (links) should be listed in a formal "Data availability" section (placed after Methods)

*** The Data Availability Section is restricted to new primary data that are part of this study. In case you have no data that require deposition in a public database, please state so instead of referring to the database: "Our study includes no data deposited in public repositories." under the heading "Data availability". ***

*** All links should resolve to a page where the data can be accessed. ***

*** Please remember to provide in the Data availability section of your revised manuscript reviewer passwords if the datasets are not yet public. ***

*** Please use detailed data citations for already available datasets that were re-analyzed in your study - for more information on the format, see point #9 below. ***

4. Please check that the title and the abstract of the manuscript are brief, yet explicit, even to non-specialists. The length of the title should not exceed 100 characters, and the abstract should be a single paragraph not exceeding 175 words.

5. All materials and methods need to be described in the manuscript using our "Structured Methods" format, which is now required for all research articles. According to this format, the Methods section includes a single "Reagents and Tools Table" - listing key reagents, experimental models, software and relevant equipment including their sources and relevant identifiers- followed by a "Methods and Protocols" section describing the methods. Please download and fill our Reagents and Tools Table template (.docx), which you can find in our author guide:

<https://www.embopress.org/page/journal/14602075/authorguide#structuredmethods>. When submitting your revised manuscript, please do not include the Reagents and Tools Table in the Methods section of the manuscript but upload it as a separate file choosing the file type "Reagent Table".

6. Please also note our reference format: <https://www.embopress.org/page/journal/14602075/authorguide#referencesformat>.

8. Please remember: digital image enhancement is acceptable practice, as long as it accurately represents the original data and conforms to community standards. If a figure has been subjected to significant electronic manipulation, this must be noted in the figure legend or in the "Materials and Methods" section. The editors reserve the right to request original versions of figures and the original images that were used to assemble the figure.

9. Our journal encourages inclusion of data citations in the reference list to directly cite datasets that were obtained from public databases. Data citations in the article text are distinct from normal bibliographical citations and should directly link to the database records from which the data can be accessed. In the main text, data citations are formatted as follows: "Data ref: Smith et al, 2001" or "Data ref: NCBI Sequence Read Archive PRJNA342805, 2017". In the Reference list, data citations must be labeled with "[DATASET]". A data reference must provide the database name, accession number/identifiers, and a resolvable link to the landing page from which the data can be accessed at the end of the reference. Further instructions are available at: <https://www.embopress.org/page/journal/14602075/authorguide#referencesformat>.

10. We request authors to consider both actual and perceived competing interests. Please review our policy (<https://www.embopress.org/page/journal/14602075/authorguide#conflictsinterest>) and update your competing interests statement if necessary. Please name this section 'Disclosure and competing interests statement' and place it after the Acknowledgements section.

11. Please note that all corresponding authors are required to provide an ORCID ID upon submission of a revised manuscript (<https://orcid.org/>). Please find instructions on how to link your ORCID ID to your account in our manuscript tracking system in our Author guidelines (<https://www.embopress.org/page/journal/14602075/authorguide#authorshipguidelines>).

12. We use CRediT to specify the contributions of each author in the journal submission system. CRediT replaces the author contribution section, which should be removed from the manuscript. Please use the free text box to provide more detailed descriptions. See also guide to authors: <https://www.embopress.org/page/journal/14602075/authorguide#authorshipguidelines>.

14. We would also welcome the submission of cover suggestions or motifs to be used by our Graphics Illustrator in designing a cover.

15. Please use the link below to submit your revision:
<https://emboj.msubmit.net/cgi-bin/main.plex>

Referee #1:

This study by López-Merino et al. presents some very interesting new findings regarding distinct roles for isoform-specific Ras GTPase signaling in neuronal synaptic plasticity. While isoform-specific Ras-ERK signaling has been heavily studied in cancer, it has not been investigated in any detail in the nervous system, despite "rasopathies" constituting a major group of neurodevelopmental disorders. In particular, here the authors employ a variety of molecular genetic and pharmacological approaches to show at synaptic connections in the hippocampus important for learning and memory that the predominant H-Ras isoform expressed in the brain is uniquely required for a prominent form of synaptic weakening known as mGluR-LTD that engages ERK signaling but is surprisingly not required for the major, canonical form of synaptic strengthening known as NMDAR-LTP which also engages ERK signaling. Accordingly, the author additionally show that H-Ras is required for important cognitive functions that utilize the hippocampus. Furthermore, the authors found that the overlooked K-Ras Isoform is instead required for NMDAR-LTP and is dispensable for mGluR-LTD. All of these findings should be of considerable interest to neuroscientists who study synaptic plasticity and neurodevelopmental disorders in particular and cell biologists who study Ras-ERK signaling in general. Overall the experiments are well-executed, the results clearly presented, and the manuscript well-written. I only have a few minor comments:

1. Are H-Ras WT and C181,184S mutant rescue constructs used in Figure 4 expressed at equal levels? Also in the Figure 4 legend in one spot the mutant is incorrectly listed as C181,185S.

2. The authors should discuss what mechanisms could be supporting the residual H-Ras-independent component of mGluR-LTD that remains with Ras-DN, H-Ras KO, Src inhibition etc. Is this the presynaptic PI3K-dependent component of mGluR-LTD previously described by the authors? Or are there other postsynaptic mGluR-LTD mechanisms still active even when H-Ras signaling is absent?

3. The authors should discuss what distinct Ras-GEFs might be involved in regulating H-Ras during mGluR-LTD vs. K-Ras during LTP? Also, are any of these possible Ras-GEFs known to be localized differentially with respect to synaptic and non-synaptic membrane like K-Ras and NMDARs vs. H-Ras and mGluR1/5?

Referee #2:

In this study by López-Merino and colleagues, the authors have investigated the the role of specific Ras isoforms in different forms of synaptic plasticity in the hippocampus (CA1). Work to date has shown a critical role for Ras in both Long term potentiation (LTP, NMDAR-LTP) and Long term depression (LTD - specifically mGluR-LTD), but the distinct roles of H-Ras and K-Ras in these forms of plasticity have not been studied. Here López-Merino et al, have used an excellent combination of functional, biochemical and molecular approaches to address this gap. They report that H-Ras is required for DHPG-LTD, but not LTP, and that K-Ras is required for LTP. Together this study is well done, the data are compelling, and the findings are both novel, interestingly and important.

I have a few comments and suggestions that I think would help to clarify some points and possibly help to shed some further light on the findings the authors have made.

- 1) In the first figure (and in other parts of the study), the authors overexpress (o/e) DN Ras or WT Ras. Is it known whether the Ras (WT or DN) are based on the H-, K- or N-Ras isoforms? It may also be worth noting that that DN Ras form used here (Ras17N) likely functions by sequestering Ras GEFs away from endogenous Ras, rather than blocking the exchange of GDP to GTP (as originally thought).
- 2) The authors refer to Erk activation - is this meant to refer to ERK1 and ERK2? Or another ERK form?
- 3) It is interesting to note that in Fig 1B, that o/e of RAS17N still increases basal levels of ERK1/2 - do the authors have any thoughts on how this may be occurring? Is it possible that this is seen because Ras17N can still bind GTP similar to the WT form?
- 4) In Figure 2A, the Ras activation kit used, similar to other Ras kits, shows a doublet band just over 20kDa - could these bands be showing different forms of Ras, or are these differentially post-translational modifications (relevant for some of the observations later in the study)? When quantifying, are both bands included in the analysis? Would it be possible to reprobe some of these blots using the K-Ras antibody to see if there is any activation of K-Ras as would be expected following cLTP stimulation, but possibly not after DHPG/NMDA stimulation?
- 5) Similar to the question above, is it known what Ras isoform is the basis for the FRas2-M probe?
- 6) The authors nicely show that blocking Src with PP2 is sufficient to block DHPG-LTD and Erk activation - does this potentially suggest that the upstream guanine nucleotide exchange factors (GEFs) involved here is likely to be SOS1? Would blocking Src with PP2 also block LTP?
- 7) In figure 4, the re-expression of WT Ras into H-Ras KO slices is a nice way to show the importance of Ras in LTD - however, it is not clear if the o/e WT Ras is H-Ras or another isoform. In the text it is referred to as either WT Ras or H-Ras. It would be very interesting to test if o/e of H-Ras vs K-Ras would have different effects on the rescue of LTD in this experiment. Also, a minor note, it is often hard to see the colours of the different symbols in the fEPSP and EPSC plots because of the error bars- is it possible to change the colour of the error bars to a grey or increase the size of the symbols to make them a little more clear?
- 8) In Figure 5A, it would be useful to label which band corresponds to H- or K-Ras in the top, Ras, blot (this also related to my comment 4). I also wonder if it would be possible to repeat this type of experiment, but with the addition of a stimulation (i.e. DHPG or cLTP treatment) to demonstrate whether H-Ras and/or K-Ras show altered sub-synaptic localisations?
- 9) is it possible to carry out an IHC with the K-Ras nKO slices to show that the knockdown is limited to only neurons - or that the mCherry is only expressed in neurons?
- 10) The use of the K-Ras nKO gives compelling evidence that this isoform is required for LTP (NMDA-LTP) and not LTD. However, it would be interesting to see if o/e of H- or K-Ras is sufficient to rescue the reduction in LTP in K-Ras nKO neurons (if this is experimentally possible). This is a similar comment to my comment 7 - i acknowledge that o/e will potentially drive unphysiological effects (necessity vs sufficiency), but this may offer greater insight into the isoform specific roles of H- vs K-Ras in LTP and LTD.
- 11) In the discussion, it may be interesting to think about whether different GEFs are engaged following either LTP or LTD, and whether this could be also driving the specific roles of H- and K-Ras in response to these plasticity stimuli. Indeed, is it known if there are GEF that preferentially bind to one isoform vs another?
- 12) It may be of use to have a schematic of H- and K-Ras protein structures (domains etc), to show how these isoforms differ etc.

Referee #3:

Small GTPase Ras proteins play central roles in synaptic transmission and plasticity, and their dysregulation is linked to various disorders. In this study, López-Merino and colleagues combined electrophysiological, live-cell imaging, biochemical, and behavioral assays to explore the synaptic functions of different Ras isoforms. They found that H-Ras is primarily localized at extrasynaptic sites and mediates a signaling cascade involving mGluR/c-Src/H-Ras/MAPK/Erk. This cascade transduces mGluR activation into mRNA translation and AMPAR internalization during mGluR-dependent LTD, distinguishing H-Ras from K-Ras, which is important for NMDAR-dependent LTP. Behavioral tests further demonstrated the critical role of H-Ras in object recognition, spatial, and social memory, underscoring its distinct function from K-Ras. These findings are novel and supported by multiple lines of independent evidence. They are important for understanding Rasopathies, particularly those caused by activating mutations in H-Ras (Costello syndrome) and K-Ras (Noonan and cardiofaciocutaneous syndromes), which are the most prevalent group of neurodevelopmental disorders. This work is a strong candidate for publication in EMBO J. I have only a minor comment to improve the study.

Minor concern:

1) Overexpressing Ras-DN and Ras-WT would affect basal transmission via both NMDAR-mediated AMPA trafficking and mGluR-mediated AMPA internalization. How these would impact plasticity measurements should be explained and discussed.

Point-by-point response to reviewers' comments

Referee #1:

This study by López-Merino et al. presents some very interesting new findings regarding distinct roles for isoform-specific Ras GTPase signaling in neuronal synaptic plasticity. While isoform-specific Ras-ERK signaling has been heavily studied in cancer, it has not been investigated in any detail in the nervous system, despite "rasopathies" constituting a major group of neurodevelopmental disorders. [...] All of these findings should be of considerable interest to neuroscientists who study synaptic plasticity and neurodevelopmental disorders in particular and cell biologists who study Ras-EK signaling in general. Overall the experiments are well-executed, the results clearly presented, and the manuscript well-written. I only have a few minor comments:

1. Are H-Ras WT and C181,184S mutant rescue constructs used in Figure 4 expressed at equal levels? Also in the Figure 4 legend in one spot the mutant is incorrectly listed as C181,185S.

In new Fig. EV4C, we now show that H-Ras WT and C181,184S mutant are expressed at equivalent levels. We have also observed that they display overall similar fluorescence intensity in infected cells (Fig. 4E). This is consistent with both constructs bearing the same promotor. We have amended the typo in Fig. 4 legend.

2. *The authors should discuss what mechanisms could be supporting the residual H-Ras-independent component of mGluR-LTD that remains with Ras-DN, H-Ras KO, Src inhibition etc. Is this the presynaptic PI3K-dependent component of mGluR-LTD previously described by the authors? Or are there other postsynaptic mGluR-LTD mechanisms still active even when H-Ras signaling is absent?*

This is an important point. Indeed, we believe that the residual component of mGluR-LTD involves a PI3K-mediated presynaptic mechanism, as we suggested in the results section (page 14, first paragraph). Nevertheless, we cannot rule out additional, Ras-independent postsynaptic mechanisms. We are now further elaborating on this point in the discussion section (page 18, lines 415-419).

3. *The authors should discuss what distinct Ras-GEFs might be involved in regulating H-Ras during mGluR-LTD vs. K-Ras during LTP? Also, are any of these possible Ras-GEFs known to be localized differentially with respect to synaptic and non-synaptic membrane like K-Ras and NMDARs vs. H-Ras and mGluR1/5?*

These are indeed very relevant points, but the experimental evidence on the role of Ras-GEFs in synaptic plasticity is very limited. Most of them come from the work of Larry Feig's laboratory. They have shown that Ras-GRF1 mediates NMDAR-dependent LTD, whereas Ras-GRF2 is involved in NMDAR-dependent LTP (Li et al 2006, J Neurosci 26(6):1721-1729). Also, while Sos2 is dispensable for LTP (Arai et al 2009, Neuroscience Letters 455 22-25), the role of Sos1 in synaptic plasticity is (to the best of our knowledge) unknown. As for their synaptic location, Ras-GRF1 and Ras-GRF2 seem to be coupled to GluN2B- (mainly extrasynaptic) and GluN2A (synaptic) NMDARs,

consistent with their roles in LTD and LTP, respectively (Li et al 2006; Krapivinsky et al. 2003, *Neuron* Nov 13;40(4):775-84). However, Ras-GRF1 and Sos1 have also been detected in the PSD (Sturani et al 1997, *Exp Cell Res* 235, 117–123; Suzuki et al 1999, *Brain Res* 840 36–44). To our knowledge, their potential role in mGluR-LTD has never been explored, probably because the role of Ras in this form of plasticity has not been addressed (until this study). We have included some of these comments in the discussion (end of page 21 and beginning of page 22).

Referee #2:

In this study by López-Merino and colleagues, the authors have investigated the role of specific Ras isoforms in different forms of synaptic plasticity in the hippocampus (CA1). [...] Here López-Merino et al, have used an excellent combination of functional, biochemical and molecular approaches to address this gap. They report that H-Ras is required for DHPG-LTD, but not LTP, and that K-Ras is required for LTP. Together this study is well done, the data are compelling, and the findings are both novel, interestingly and important.

I have a few comments and suggestions that I think would help to clarify some points and possibly help to shed some further light on the findings the authors have made.

1) In the first figure (and in other parts of the study), the authors overexpress (o/e) DN Ras or WT Ras. Is it known whether the Ras (WT or DN) are based on the H-, K- or N-Ras isoforms? It may also be worth noting that that DN Ras form used here (Ras17N) likely functions by sequestering Ras GEFs away from endogenous Ras, rather than blocking the exchange of GDP to GTP (as originally thought).

We now indicate in Methods (page 30, lines 703-705) and in the Results section (page 6, line 101) that Ras WT and DN forms were made from H-Ras sequence. We agree with the reviewer that the mechanism of action of Ras DN likely involves sequestering of Ras GEFs, probably because of its increase affinity for GDP, as reported in the literature (Feig & Cooper, *Mol Cell Biol* 1988 Aug;8(8):3235-43; Nassar et al, *Biochemistry* 2010 Mar 9;49(9):1970-4). We have included a sentence in the text (page 21, lines 480-486/523) to discuss this issue, and how it may be linked to the unspecific effects of Ras DN by interfering with other Ras isoforms (i.e. K-Ras in LTP) (Matallanas et al, *J. Biol. Chem.* 278: 4572–4581, 2003).

2) The authors refer to Erk activation - is this meant to refer to ERK1 and ERK2? Or another ERK form?

Yes, we are referring to ERK1/2. We have made changes in our figures and throughout the text to include this information.

3) It is interesting to note that in Fig 1B, that o/e of RAS17N still increases basal levels of ERK1/2 - do the authors have any thoughts on how this may be occurring? Is it possible that this is seen because Ras17N can still bind GTP similar to the WT form?

We must note that, under basal conditions, pERK1/2 levels were not significantly different between Ras-DN and GFP conditions. We have included these statistical

comparisons in the new Fig. 1B. However, we have observed that infection *per se* tends to increase basal ERK activity, as compared to uninfected slices. Yet, this effect was not statistically significant, and occurred for both GFP and Ras-DN. We have made changes in the text (page 7, lines 134-142) to clarify this point. In contrast, our data clearly indicates that activity-dependent (DHPG-induced) ERK1/2 activation is mediated by Ras (blocked by Ras-DN; Fig 1B).

4) *In Figure 2A, the Ras activation kit used, similar to other Ras kits, shows a doublet band just over 20kDa - could these bands be showing different forms of Ras, or are these differentially post-translational modifications (relevant for some of the observations later in the study)? When quantifying, are both bands included in the analysis? WOULD it be possible to reprobe some of these blots using the K-Ras antibody to see if there is any activation of K-Ras as would be expected following cLTP stimulation, but possibly not after DHPG/NMDA stimulation?*

We are quite certain that the lower band corresponds to HRas, as it virtually disappears in HRas KO (Fig 5A and new Figs. EV4 and EV5). However, we cannot conclude that the upper band belongs exclusively to K-Ras. This is because i) DHPG causes an increase in both bands (GTP-bound) in WT slices, and ii) this effect is completely blocked in H-Ras KO slices (new Fig. EV4). Therefore, it is possible that some signal from the upper band corresponds to post-translational modifications of H-Ras that is activated by DHPG. In any event, unless otherwise stated, both bands were quantified together, as we indicate now in the legend of Fig. 2A. Additionally, as pointed out by the reviewer, we have performed new experiments showing that K-Ras is activated following cLTP induction (new Fig. EV5E). And indeed, the fact that DHPG-induced Ras activation was fully suppressed in H-Ras KO slices (new Fig. EV4) indicates that K-Ras activity is not affected by DHPG. We have discussed these issues in our revised manuscript (page 13, lines 277-280; page 17, lines 375-376).

5) *Similar to the question above, is it known what Ras isoform is the basis for the FRas2-M probe?*

Yes, we are using HRas-GFP for FRET experiments. We have included this information in the Methods section (page 33, line 789) and in the legends of Fig. 2 and EV2.

6) *The authors nicely show that blocking Src with PP2 is sufficient to block DHPG-LTD and Erk activation - does this potentially suggest that the upstream guanine nucleotide exchange factors (GEFs) involved here is likely to be SOS1? Would blocking Src with PP2 also block LTP?*

This is an interesting point. There is little work published on the role of Sos1 in synaptic plasticity, despite being known to be present at synapses for many years now (Suzuki et al 1999, Brain Res 840 36–44). It is known that Sos2 is dispensable for LTP (Arai et al 2009, Neuroscience Letters 455 22–25), but we have not found studies in the literature linking Sos1 with specific synaptic plasticity paradigms. RasGRF1/2 have been related to NMDAR-dependent LTD/LTP, respectively (Li et al 2006, J Neurosci 26(6):1721–1729). However, which Ras-GEFs are involved in mGluR-LTD still remains to be addressed. Regarding the role of Src in LTP, a recent paper reports that Src regulation of LTP is

dependent on specific GluN1 splice variants (Li et al 2024; Philos Trans R Soc Lond B Biol Sci. 2024). We have included some of these comments in the discussion (end of page 21 and first paragraph of page 22).

7) *In figure 4, the re-expression of WTRas into H-Ras KO slices is a nice way to show the importance of Ras in LTD - however, it is not clear if the o/e WTRas is H-Ras or another isoform. In the text it is referred to as either WTRas or H-Ras. It would be very interesting to test if o/e of H-Ras vs K-Ras would have different effects on the rescue of LTD in this experiment. Also, a minor note, it is often hard to see the colours of the different symbols in the fEPSP and EPSC plots because of the error bars- is it possible to change the colour of the error bars to a grey or increase the size of the symbols to make them a little more clear?*

We now explicitly indicate in the text (page 13, line 288) that we are using H-Ras WT for our rescue experiments in Fig. 4D. As for the rescue experiment with K-Ras, in our opinion, this experiment (no matter the outcome) would not provide further insight for the role of specific Ras isoforms in synaptic plasticity. This is because the H-Ras KO already has endogenous K-Ras present (Fig. 5A). Therefore, the overexpression of recombinant K-Ras is not expected to rescue mGluR-LTD (as endogenous K-Ras did not). And if it does, it would be the result of unspecific effects (i.e. protein mistargeting to not physiological locations or else) due to overexpression. In fact, the reviewer acknowledges this interpretation later in point 10.

As requested by the reviewer, we have increased the size of the symbols in the graphs from Fig.4D. We believe the data now looks clearer.

8) *In Figure 5A, it would be useful to label which band corresponds to H- or K-Ras in the top, Ras, blot (this also related to my comment 4). I also wonder if it would be possible to repeat this type of experiment, but with the addition of a stimulation (i.e. DHPG or cLTP treatment) to demonstrate whether H-Ras and/or K-Ras show altered sub-synaptic localisations?*

As indicated in response to point 4, we are not certain about the nature of the two Ras bands, particularly for the upper band, which may contain K-Ras and H-Ras contributions. Therefore, we decided not to label them in our blots.

On the other hand, we have performed the requested experiment and carried out membrane fractionations from WT and H-Ras KO acute slices following DHPG treatment. The results are now shown in new Fig. EV5A. There was no significant redistribution of H-Ras after stimulation, but interestingly, K-Ras presence in the PSD fraction was decreased upon DHPG treatment (page 15, lines 337-339).

9) *is it possible to carry out an IHC with the K-Ras nKO slices to show that the knockdown is limited to only neurons - or that the mCherry is only expressed in neurons?*

To address this point, we now provide high magnification images of CA1 pyramidal neurons expressing mCherry (from the AVV-CaMKII α -mCherry-Cre virus) and DAPI in separate channels (new Fig. EV5B). The exclusive expression of Cre-mCherry in pyramidal neurons with AVV-CaMKII α -mCherry-Cre was also confirmed by immunohistochemistry with NeuN in a previous study of the laboratory (Nat Comm. 2019 Jul 4;10(1):2968; Suppl Fig. 12).

10) *The use of the K-Ras nKO gives compelling evidence that this isoform is required for LTP (NMDA-LTP) and not LTD. However, it would be interesting to see if o/e of H- or K-Ras is sufficient to rescue the reduction in LTP in K-Ras nKO neurons (if this is experimentally possible). This is a similar comment to my comment 7 - i acknowledge that o/e will potentially drive unphysiological effects (necessity vs sufficiency), but this may offer greater insight into the isoform specific roles of H- vs K-Ras in LTP and LTD.*

Following the same rational as in response to point 7, we believe the overexpression of H-Ras in a K-Ras KO background, where endogenous H-Ras is preserved (Fig. EV5C, total Ras blot) may render similar uncertain conclusions, as any potential effect would be due to overexpression. Just to clarify, the rescue experiment we did carry out with the full (not conditional) H-Ras KO (shown in Fig. 4D) was designed to address whether mGluR-LTD deficits were due to developmental defects or to the absence of H-Ras in CA1 neurons (or alternatively in other cell types). In contrast, in the case of LTP deficits following acute K-Ras KO in hippocampal pyramidal neurons, the developmental defects are excluded, and so it is the potential contribution to LTP of K-Ras expression in other cell types (glia, interneurons, etc). Therefore, we believe that in this case, the K-Ras rescue experiment will not provide additional evidence to the role of K-Ras in LTP.

11) *In the discussion, it may be interesting to think about whether different GEFs are engaged following either LTP or LTD, and whether this could be also driving the specific roles of H- and K-Ras in response to these plasticity stimuli. Indeed, is it known if there are GEF that preferentially bind to one isoform vs another?*

As we have argued in response to point 6, there is compelling evidence for the role of Ras-GRF2 in NMDA-LTP (Li et al 2006; Schwechter et al 2013, PNAS 110 (35), 14462-14467). In contrast, little is known about which Ras-GEFs are involved in mGluR-LTD, likely due to the fact that the role of Ras itself in this form of plasticity has not been directly demonstrated until this study. To the best of our knowledge, there are no studies showing preferential binding of Ras-GEF for specific Ras isoforms in neurons. Nevertheless, Ras-GRF1 and Sos1 seem to pair up better with H-Ras in other cell types (Matallanas et al 2003; J Biol Chem.;278:4572-81; Jaumot et al 2002, J Biol Chem 277:272-8). We have included some of these comments in the discussion (end of page 21, first paragraph of page 22).

12) *It may be of use to have a schematic of H- and K-Ras protein structures (domains etc), to show how these isoforms differ etc.*

As suggested, we have included a scheme of H- and K-Ras primary structures in new Fig. 4F.

Referee #3:

Small GTPase Ras proteins play central roles in synaptic transmission and plasticity, and their dysregulation is linked to various disorders. In this study, López-Merino and colleagues

combined electrophysiological, live-cell imaging, biochemical, and behavioral assays to explore the synaptic functions of different Ras isoforms. [...] These findings are novel and supported by multiple lines of independent evidence. They are important for understanding Rasopathies, particularly those caused by activating mutations in H-Ras (Costello syndrome) and K-Ras (Noonan and cardiofaciocutaneous syndromes), which are the most prevalent group of neurodevelopmental disorders. This work is a strong candidate for publication in EMBO J. I have only a minor comment to improve the study.

Minor concern:

1) Overexpressing Ras-DN and Ras-WT would affect basal transmission via both NMDAR-mediated AMPA trafficking and mGluR-mediated AMPA internalization. How these would impact plasticity measurements should be explained and discussed.

Our electrophysiology data indicate that overexpression of either Ras-DN or Ras-WT did not affect AMPA nor NMDA currents (Fig. EV1). However, previous work from Julius Zhu and colleagues has shown that Ras-DN impairs LTP and depresses AMPA receptor currents in an activity-dependent manner (Zhu et al 2002, Cell 110: 443–455). On the one hand, the effect of H-Ras-DN on LTP and activity-dependent AMPAR synaptic delivery may be explained by the unspecific targeting of H-Ras-DN to synapses, where it could interfere with endogenous K-Ras and thereby impair LTP (discussed before for referee #2, point 1; see also Matallanas et al, J. Biol. Chem. 278: 4572–4581, 2003). We are now elaborating on these issues in our revised manuscript (page 21, lines 480-486). On the other hand, the fact that we did not observe an effect of H-Ras-DN on basal AMPAR transmission may be due to different levels of spontaneous activity in our organotypic slice cultures as compared to the ones used in the study by Zhu et al.

As for the effects on mGluR-dependent AMPA internalization, we believe that Ras/Erk signaling triggers Arc translation which causes AMPA receptor internalization, as previously described (Waung et al., 2008, Neuron 10;59(1):84-97). Thus, Ras-DN would prevent AMPA receptor internalization by blocking MAPK activation (Fig. 1B) and subsequent *de novo* protein synthesis (Fig. 1C). These points have also been included in the discussion (end of page 18, first paragraph of page 19).

Dear Dr. Briz,

Thank you for submitting your revised manuscript (EMBOJ-2024-118427R) to The EMBO Journal for our consideration. It has now been seen by two of the original referees who previously assessed the initial version of your manuscript (their comments are included below). I am glad to say that both referees are very satisfied with the revised manuscript, point out that the previously raised concerns have been successfully addressed, and now support the publication of this interesting manuscript in The EMBO Journal without any further comments. Given this input, I am glad to let you know that your manuscript has in principle been accepted for publication in our journal.

Before we can proceed with the production of your article, there are a few editorial requests/formatting changes that we need from you to address in the final version of your manuscript:

- Please note that no more than 5 keywords can be listed after the Abstract (you currently list 10).
- Please change the heading of your conflict-of-interest statement (currently: "Declaration of interests") to "Disclosure and competing interests statement".
- During our standard Figure checks, we detected reuse of blots between Figure 5A and Figure EV3A that is not listed in the Figure legends. Please make sure that the reuse is intentional and justified by the experimental setup, and -if so- please detail the reuse in the Figure legend.
- During our routine pre-acceptance checks, our data editors have raised the following queries regarding figures, data, and legends. Please make sure that all requests below are completely addressed in the final version of your manuscript:
 1. Please provide the exact p values in the legends of Figures 1A, C; 2A, B; 3D, F; 4B, C, E; 5C, D.
 2. Please note that information related to "n" is missing in the legends of Figures 1A, EV2.
 3. Please note that the error bars must be defined in the legends of Figures 4B, E; EV3 D.
 4. Please note that the scale bar needs to be defined in the legend of Figure 4E.
 5. Please note that the white arrow heads are not defined in the legend of Figure EV2. This needs to be rectified.
- We also kindly request you to change the order of the manuscript sections as follows: Title page - Abstract & Keywords - Introduction - Results - Discussion - Methods - Data Availability - Acknowledgements - Disclosure and Competing Interests Statement - References - Figure Legends - main Tables (if there are any) - Expanded View Figure Legends.

Please also note that as part of the EMBO publications' Transparent Editorial Process, The EMBO Journal publishes online a Peer Review File along with each accepted manuscript. This File will be published in conjunction with your paper and will include the referee reports, your point-by-point response and all pertinent correspondence relating to the manuscript. You can opt out of this by letting the editorial office know (contact@embojournal.org). If you do opt out, the Peer Review File link will point to the following statement: "No Peer Review File is available with this article, as the authors have chosen not to make the review process public in this case."

We look forward to seeing a final version of your manuscript as soon as possible. Please let us know if you have any questions and use this link to submit your revision: <https://emboj.msubmit.net/cgi-bin/main.plex>.

Best wishes,

Ioannis

Referee #1:

The authors have done an excellent job of addressing the prior reviewer comments, both by adding new experimental data and by adding more requested information and relevant discussion in the text. I recommend publication of this very interesting and rigorous study without further delay.

Referee #2:

The Authors have more than addressed my questions - this is a very interesting paper, and I look forward to seeing it published.

All editorial and formatting issues were resolved by the authors.

Dear Dr. Briz,

Congratulations on an excellent manuscript! I am very pleased to inform you that it has been accepted for publication in The EMBO Journal. Thank you very much for your comprehensive responses to the referees' comments, and for addressing all editorial and formatting requests.

If you have any questions, please do not hesitate to contact the Editorial Office. Thank you for your contribution to The EMBO Journal. Working with you has been a pleasure!

Best regards,

Ioannis
